# PRE-TRAINING FOR ROBOTS: LEVERAGING DIVERSE MULTITASK DATA VIA OFFLINE RL

## ABSTRACT

Recent progress in deep learning highlights the tremendous potential of utilizing diverse datasets for achieving effective generalization and makes it enticing to consider leveraging broad datasets for attaining more robust generalization in robotic learning as well. However, in practice we likely will want to learn a new skill in a new environment that is unlikely to be contained in the prior data. Therefore we ask: how can we leverage existing diverse offline datasets in combination with small amounts of task-specific data to solve new tasks, while still enjoying the generalization benefits of training on large amounts of data? In this paper, we demonstrate that end-to-end offline RL can be an effective approach for doing this, without the need for any representation learning or vision-based pre-training. We present pre-training for robots (PTR), a framework based on offline RL that attempts to effectively learn new tasks by combining pre-training on existing robotic datasets with rapid fine-tuning on a new task, with as a few as 10 demonstrations. At its core, PTR applies an existing offline RL method such as conservative Q-learning (CQL), but extends it to include several crucial design decisions that enable PTR to actually work and outperform a variety of prior methods. To the best of our knowledge, PTR is the first offline RL method that succeeds at learning new tasks in a new domain on a real WidowX robot with as few as 10 task demonstrations, by effectively leveraging an existing dataset of diverse multi-task robot data collected in a variety of toy kitchens. We present an accompanying overview video at this Anonymous URl: https://www.youtube.com/watch?v=yAWgyLJD5lY&ab_channel=PTRICLR.

## 1 INTRODUCTION

Robotic learning methods based on reinforcement learning (RL) or imitation learning (IL) have led to a number of impressive results (Levine et al., 2016; Kalashnikov et al., 2018; Young et al., 2020; Kalashnikov et al., 2021; Ahn et al., 2022), but the generalization abilities of policies learned in this way are typically limited by the quantity and breadth of the data available to train them. In practice, the cost of real-world data collection for each task means that such methods often use smaller datasets, which leads to more limited generalization. A natural way to circumvent this limitation is to incorporate existing diverse robotic datasets into the training pipeline of a robot learning algorithm, analogously to how pretraining on diverse prior datasets has enabled rapid finetuning in supervised learning fields, such as computer vision and NLP. But how can we devise algorithms that enable effective pretraining for robotic RL?

In most cases, answering this question requires a method that can pre-train on existing data from a wide range of tasks and domains, and then provide a good starting point for efficiently learning a *new* task in a *new* domain. Prior approaches utilize such existing data by running imitation learning (IL) (Young et al., 2020; Ebert et al., 2021; Shafiullah et al., 2022) or by using representation learning (Nair et al., 2022) methods for pre-training and then fine-tuning with imitation learning. However, this may not necessarily lead to representations that can reason about the consequences of their actions. In contrast, end-to-end RL can offer a more general paradigm, that can be effective for both pre-training and fine-tuning, and is applicable even when assumptions in prior work are violated. Therefore we ask, can we devise a simple and unified framework where *both* the pretraining and finetuning process uses RL? This presents significant challenges pertaining to leveraging large amounts of offline multi-task datasets, which would require high capacity models and this can be very challenging (Bjorck et al., 2021).

In this paper, we show that multi-task offline RL pretraining on diverse multi-task demonstration data followed by offline RL finetuning on a very small number of trajectories (as few as 10 trials, a maximum of 15) can indeed be made into an effective robotic learning strategy that in practice can significantly outperform methods based on

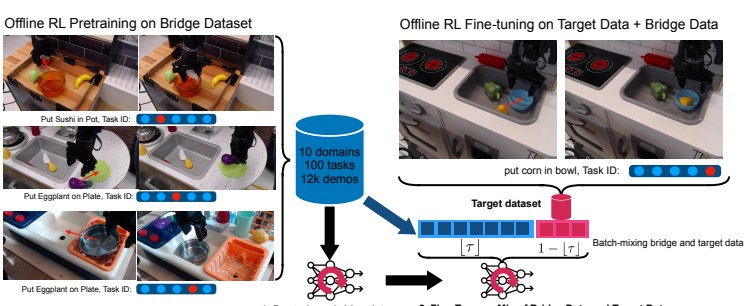

Figure 1: **Overview of PTR:** We first perform general offline pre-training on diverse multi-task robot data and subsequently finetune on one or several a target tasks while mixing batches between the prior data and the target dataset using a batch mixing ratio of $\tau$.

imitation learning pre-training as well as RL-based methods that do not employ pre-training. This is surprising and significant, since prior work (Mandlekar et al., 2021) has claimed that IL methods are superior to offline RL when provided with human demonstrations. Our framework, which we call PTR (pre-training for robots), is based on the CQL algorithm (Kumar et al., 2020), but introduces a number of design decisions that we show are critical for good performance and enable large-scale pre-training. These choices include a specific choice of architecture for providing high capacity while preserving spatial information, the use of group normalization, and an approach for feeding actions into the model that ensures that actions are used properly for value prediction. We experimentally validate these design decisions and show that PTR benefits from increasing the network capacity, even with large ResNet50 architectures, which have never been previously shown to work with offline RL. Our experiments utilize the bridge dataset (Ebert et al., 2021), which is an extensive previously collected dataset consisting of thousands of trials for a very large number of robotic manipulation tasks in multiple environments. A schematic of our framework is shown in Figure 1.

The main contribution of this work is a demonstration that PTR can enable offline RL pre-training on diverse real-world robotic data, and that these pre-trained policies can be fine-tuned to learn new tasks with just 10-15 demonstrations. This is a significant improvement over prior RL-based pre-training and finetuning methods, which typically require hundreds or even thousands of trials (Singh et al., 2020; Kalashnikov et al., 2021; Julian et al., 2020; Chebotar et al., 2021; Lee et al., 2022a). We present a detailed analysis of the design decisions that enable offline RL to provide an effective pretraining framework, and show empirically that these design decisions are crucial for good performance. Although the individual components that constitute PTR are based closely on prior work, we show that the novel combination of these components in PTR is important to make offline RL into a viable pre-training tool that can outperform other pre-training approaches and other RL-based policy learning strategies.

## 2 RELATED WORK

A number of prior works have proposed algorithms for offline RL (Fujimoto et al., 2018; Kumar et al., 2019; 2020; Kostrikov et al., 2021a;b; Wu et al., 2019; Jaques et al., 2019; Fujimoto & Gu, 2021; Siegel et al., 2020). Especially, many prior works study offline RL with multi-task data and devise techniques that perform parameter sharing(Wilson et al., 2007; Parisotto et al., 2015; Teh et al., 2017; Espeholt et al., 2018; Hessel et al., 2019), or perform data sharing or relabeling (Yu et al., 2021; Andrychowicz et al., 2017; Yu et al., 2022; Kalashnikov et al., 2021; Xie & Finn, 2021). In this paper, our goal is not to develop new offline RL algorithms, but to show that these offline RL algorithms can be an effective tool to pre-train from prior data and then fine-tune to new tasks, and we illustrate the design decisions required to get such methods to work well.

Unlike methods for fine-tuning from a learned initialization (Nair et al., 2020; Kostrikov et al., 2021b; Lee et al., 2022c), which typically perform online interaction, we consider the setting where we do not use any online interaction and do not require access to a reward function. This resembles the problem setting considered by offline meta-RL methods (Li et al., 2019; Dorfman & Tamar, 2020; Mitchell et al., 2021; Pong et al., 2021; Lin et al., 2022). However, our approach is simpler as we simply fine-tune the very same offline RL algorithm and our method, PTR, outperforms the meta-RL method from Mitchell et al. (2021). Xu et al. (2022) present a prompting-based few-shot adaptation approach based on decision-transformers that is also related to our work.

In our experiments, we compare our approach to other approaches that attempt to leverage large, diverse datasets via representation learning (Mandlekar et al., 2020; Yang & Nachum, 2021; Yang et al., 2021; Nair et al., 2022; He et al., 2021), as well as other methods for learning from human demonstrations, such as behavioral cloning methods with expressive policy architectures (Shafiullah et al., 2022). We find that PTR generally outperforms these methods. We also perform an empirical analysis to identify the design decisions behind the improved performance of RL-based PTR on demonstration data compared to BC, and find that the gains largely come from the efficacy of the value function in identifying the most "critical" decisions in a trajectory. This insight is in contrast to prior work (Mandlekar et al., 2021), which argues that offline RL methods often do not work on real robots with human demonstration data, and supports the analysis in prior work (Kumar et al., 2022) that aims to understand when offline RL outperforms BC with demonstration data.

Perhaps the most closely related to our work are prior methods that run model-free offline RL on diverse real-world data and then finetune to new tasks (Singh et al., 2020; Kalashnikov et al., 2021; Julian et al., 2020; Chebotar et al., 2021; Lee et al., 2022a). These prior methods typically perform *online* fine-tuning, which requires complex reset mechanisms and uses *significantly* more data. (Chebotar et al., 2021; Kalashnikov et al., 2021) require a thousand trials for a new task, and (Julian et al., 2020; Lee et al., 2022a) require multiple hours or even days of online collection to learn new tasks. While these prior methods are autonomous, our framework PTR performs offline fine-tuning to learn a new task with as few as 10 demonstrations and thus provides a complementary advantage.

## 3 PRELIMINARIES AND PROBLEM STATEMENT

RL methods are derived under the formal model of a Markov decision process (MDP), which is a tuple $\mathcal{M} = (\mathcal{S}, \mathcal{A}, T, r, \mu_0, \gamma)$, where $\mathcal{S}, \mathcal{A}$ denote the state and action spaces, and $T(\mathbf{s}'|\mathbf{s}, \mathbf{a})$, $r(\mathbf{s}, \mathbf{a})$ represent the dynamics and reward function respectively. $\mu_0(\mathbf{s})$ denotes the initial state distribution, and $\gamma \in (0, 1)$ denotes the discount factor. The policy $\pi(\mathbf{a}|\mathbf{s})$ learned by RL agents must optimize the long-term cumulative reward, $\max_\pi J(\pi) := \mathbb{E}_{(\mathbf{s}_t, \mathbf{a}_t) \sim \pi}[\sum_t \gamma^t r(\mathbf{s}_t, \mathbf{a}_t)]$.

**Problem statement.** Our goal is to learn general-purpose initializations from a broad, multi-task offline dataset and then finetune these initializations to specific downstream tasks. We denote the general-purpose offline dataset by $\mathcal{D}$, which is partitioned into $k$ chunks. Each chunk contains data for a given robotic task (e.g. picking and placing a given object) collected in a given domain (e.g. a particular kitchen). See Figure 1 for an illustration. Denoting the task/domain abstractly using an identifier $i$, the dataset can be formally represented as $\mathcal{D} = \cup_{i=1}^k (i, \mathcal{D}_i)$, where we denote the set of training tasks concisely as $\mathcal{T}_{\text{train}} = [k]$. Chunk $\mathcal{D}_i$ consists of data for a given task identifier $i$, and consists of a collection of transition tuples, $\mathcal{D}_i = \{(\mathbf{s}_j^i, \mathbf{a}_j^i, r_j^i, \mathbf{s}_j'^i)\}_{j=1}^n$ collected by a demonstrator on task $i$. Our goal is to utilize this multi-task dataset $\mathcal{D}$, to find the best possible policy for one or multiple target tasks (denoted without loss of generality as task $\mathcal{T}_{\text{target}} = \{k+1, \cdots, n\}$), for which no experience is observed in $\mathcal{D}$. While the diverse dataset $\mathcal{D}$ does not contain any experience for the target tasks, we are provided with a very small dataset of demonstrations $\mathcal{D}^* := \{\mathcal{D}_{k+1}^*, \mathcal{D}_{k+1}^*, \cdots, \mathcal{D}_n^*\}$ corresponding to each of the target tasks. Note that the size of $\mathcal{D}^*$ is extremely small: in our experiments we consider between 10 to 15 demonstrations for a given target task, such that a policy that simply ignores the diverse offline dataset is unlikely to succeed. Our goal is to attain the best possible policy for tasks $\mathcal{T}_{\text{target}}$ at the end.

**Background and preliminaries.** The Q-value of a given state-action tuple $Q^\pi(\mathbf{s}, \mathbf{a})$ for a policy $\pi$ is the long-term discounted reward attained by executing action $\mathbf{a}$ at state $\mathbf{s}$ and following policy $\pi$ thereafter. The Q-function satisfies the Bellman equation $Q^\pi(\mathbf{s}, \mathbf{a}) = r(\mathbf{s}, \mathbf{a}) + \gamma \mathbb{E}_{\mathbf{s}', \mathbf{a}'}[Q^\pi(\mathbf{s}', \mathbf{a}')]$. Typical model-free offline RL methods (Fujimoto et al., 2018; Kumar et al., 2019; 2020) alternate between estimating the Q-function of a fixed policy $\pi$ using the offline dataset $\mathcal{D}$ and then improving the policy $\pi$ to maximize the learned Q-function. Our system, PTR, utilizes one such model-free offline-RL method, conservative Q-learning (CQL) (Kumar et al., 2020). We discuss how we adapt CQL for pre-training on diverse data followed by single-task finetuning in Section 4.

**Tasks and domains**. As discussed, our problem involves pre-training on data from many tasks and domains, which we source from the bridge dataset Ebert et al. (2021), and finetuning to a new task in a new domain. Our terminology for "task" and "domain" follows Ebert et al. (2021): a task corresponds to a skill-object pair, such as "put potato in pot" and a domain corresponds to a particular environment, which in the case of the bridge dataset consists of different toy kitchens, potentially with different viewpoints and robot placements. We assume the new tasks and environments come from the same training distribution, but are not seen in the prior data.

# 4 LEARNING POLICIES FOR NEW TASKS FROM OFFLINE RL PRE-TRAINING

To effectively solve new tasks from diverse offline datasets, a robotic learning framework must: **(1)** extract useful skills out of the diverse robotic dataset, and **(2)** rapidly specialize the learned skills towards an unseen target task, given only a minimal amount of experience from this target task. In this section, we present our framework, PTR, that provides these benefits by training a single, highly-expressive deep neural via offline RL, and then specializes it on the target task with a small amount of data. We will first present the key components of our robotic framework in Section 4.1 and then discuss our novel technical contributions, the practical design choices that are crucial for attaining good performance in Section 4.2.

## 4.1 THE COMPONENTS OF PTR

To satisfy both requirements **(1)** and **(2)** from above, our framework uses a multi-task offline RL approach, where the policy and Q-function are conditioned on a task identifier. This allows us to share a single set of weights for all possible tasks in the diverse offline dataset, providing a general-purpose pre-training procedure that can use diverse data. Once a policy is obtained via this multi-task pre-training process, we adapt this policy for solving a new target task by utilizing a very small amount of target task data. We describe the two phases, pre-training and fine-tuning, below:

**Phase 1: Multi-task offline RL pre-training.** In the first phase, PTR learns a single Q-function and policy for all tasks $i \in \mathcal{T}_{\text{train}}$ conditioned on the task identifier $i$, i.e., $Q_\phi(\mathbf{s}, \mathbf{a}; i)$ and $\pi_\theta(\mathbf{a}|\mathbf{s}, i)$, via multi-task offline RL. We use a one-hot task identifier that imposes minimal assumptions on the task structure. For multi-task offline RL, we use the conservative Q-learning (CQL) (Kumar et al., 2020) algorithm, extending it to the multi-task setting. This amounts to training the multi-task Q-function against a temporal difference error objective along with a regularizer that explicitly minimizes the expected Q-value under the learned policy $\pi_\theta(\mathbf{a}|\mathbf{s}; i)$, to prevent overestimation of Q-values for unseen actions, which can lead to poor offline RL performance (Kumar et al., 2019). Formally, the training objective for our multi-task Q-function, as prescribed by CQL, is given by:

$$\min_\phi \ \alpha \left( \underset{\substack{i \sim \mathcal{T}_{\text{train}}, \\ \mathbf{s} \sim \mathcal{D}_i, \mathbf{a} \sim \pi}}{\mathbb{E}} [Q_\phi(\mathbf{s}, \mathbf{a}; i)] - \underset{\substack{i \sim \mathcal{T}_{\text{train}}, \\ \mathbf{s}, \mathbf{a} \sim \mathcal{D}}}{\mathbb{E}} [Q_\phi(\mathbf{s}, \mathbf{a}; i)] \right) + \frac{1}{2} \underset{\substack{i \sim \mathcal{T}_{\text{train}}, \\ \mathbf{s}, \mathbf{a}, \mathbf{s}' \sim \mathcal{D} \\ \mathbf{a}' \sim \pi}}{\mathbb{E}} \left[ \left( Q_\theta(\mathbf{s}, \mathbf{a}; i) - r - \gamma \bar{Q}(\mathbf{s}', \mathbf{a}') \right)^2 \right],$$

$\bar{Q}$ denotes a target Q-network, which a delayed copy of the current Q-network. We train $\phi$ by running gradient descent on the above objective, and then optimize the learned policy to maximize the learned Q-values, along with an additional entropy regularizer as shown below:

$$\max_\theta \quad \mathbb{E}_{i \sim \mathcal{T}_{\text{train}}, \mathbf{s} \sim \mathcal{D}_i} \left[ \mathbb{E}_{\mathbf{a} \sim \pi_\theta(\cdot|\mathbf{s}; i)} [Q_\phi(\mathbf{s}, \mathbf{a}; i)] \right] + \beta \mathcal{H}(\pi_\theta).$$

At the end of this multi-task offline training phase, we obtain a policy $\pi_\theta^{\text{off}}$ and Q-function $Q_\phi^{\text{off}}$, that are ready to be finetuned to a new downstream task.

**Phase 2: Offline fine-tuning of $\pi_\theta^{\text{off}}$ and $Q_\phi^{\text{off}}$ to target tasks $\mathcal{T}_{\text{target}}$.** In the second phase, PTR attempts to learn a policy to solve one or more downstream tasks by adapting $\pi_\theta^{\text{off}}$, using a limited set of user-provided demonstrations that we denote $\mathcal{D}^*$. Our method for adaptation is simple yet effective: we incorporate the new target task data into the replay buffer of the very same offline multi-task CQL algorithm from the previous phase and resume training from Phase 1. However, naïvely incorporating the target task data into the replay buffer might still not be effective since this scheme would hardly ever train on the target task data during adaptation due to the large imbalance between the sizes of the few target demonstrations and the large pre-training dataset. To address this imbalance, each minibatch passed to multi-task CQL during offline fine-tuning consists of a $\tau$ fraction of transitions from bridge demonstration data and $1 - \tau$ fraction of transitions from the target dataset. By setting $\tau$ to be small, we are able to prioritize multi-task CQL to look at target task data frequently, enabling it to make progress on the downstream task without overfitting.

**Handling task identifiers for new tasks.** The description of our system so far has assumed that the downstream test tasks are identified via a task-identifier. In practice, we utilize a one-hot vector to indicate the index of a task. While such a scheme is simple to implement, it is not quite obvious how we should incorporate new tasks with one-hot task identifiers. In our experiments, we use two approaches for solving this problem: first, we can utilize a larger one-hot encoding that incorporates tasks in both $\mathcal{T}_{\text{train}}$ and $\mathcal{T}_{\text{target}}$, but never train the network corresponding to $\mathcal{T}_{\text{target}}$. The Q-function and the policy are trained on these *placeholder* task identifiers only during fine-tuning in Phase 2.

Another approach for handling new tasks is to not use unique task identifiers for every new task, but rather "*re-target*" or re-purpose existing task identifiers for new target tasks in the fine-tuning phase. PTR provides the option: we can simply assign an already existing task identifier to the target demonstration data before fine-tuning the learned Q-function and the policy. For example, in our experiments in Section 5 we re-target the put sushi in pot task which uses orange transparent pots to instead put the sushi into a metal pot, which was never seen during training.

A complete overview of our approach is shown in Figure 1. We use a value of $\alpha = 10.0$ in multi-task CQL and $\tau = 0.8$ for mixing the pre-training dataset and the target task dataset in most of our experiments in the real-world, without requiring any domain-specific tuning.

## 4.2 Important Design Choices and Practical Considerations

Even though the components discussed in Section 4.1 are sufficient to give rise to an offline pre-training and fine-tuning algorithm, as we show in Section 4, this approach does not lead very good results just on its own. Instead, we must make some crucial design decisions, including designing neural network architectures that can learn from diverse data, cross-validation metrics to identify policies we expect to be effective after fine-tuning, and the design of the reward functions that can be used to label the pre-training dataset. We will show that making the right choices for these components leads to significant improvement (more than **3.5x** in final real-world performance). Thus, describing, analyzing, and evaluating these design choices is an important part of this work that we hope will facilitate real-world applications of offline RL pretraining.

**Policy and Q-function architectures.** Perhaps the most crucial design decision for our approach is the neural network architecture for representing $\pi^{\text{off}}$ and $Q^{\text{off}}$. Since we wish to fine-tune the policy for different tasks, we must use high-capacity neural network models for representing the policy and the Q-function. We experimented with a variety of standard (high-capacity) architectures for vision-based robotic RL. This includes standard convolutional architectures (Singh et al., 2020) and IMPALA architectures (Espeholt et al., 2018). However, we observed that these standard models were unable to effectively handle the diversity of the pre-training data, and often collapsed. Then,

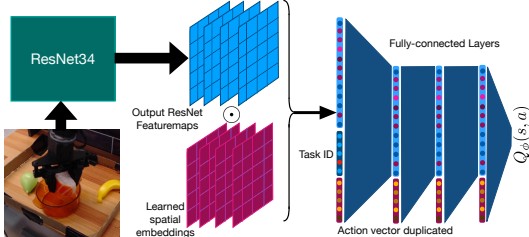

Figure 2: **The Q-function architecture for PTR.** The encoder is a ResNet34 with group normalization along with learned spatial embeddings (left). The decoder (right) is a MLP with the action vector duplicated and passed in at each layer. A one-hot task identifier is also passed into the input of the decoder.

we attempted to utilize standard ResNets (He et al., 2016) (ResNet-18, Resnet-34, and their adaptations to imitation problems from Ebert et al. (2021)) to represent $Q_\phi$, but faced divergence challenges similar to prior efforts with batch normalization (Bjorck et al., 2021; Bhatt et al., 2019). We found that by simply replacing batch normalization layers – known to be hard to train with TD-learning (Bhatt et al., 2019) – with **group normalization** layers (Wu & He, 2018), we were able to stably train ResNet Q-functions. See Appendix E for quantitative studies comparing these choices. Unlike prior work (Lee et al., 2022b), we observed that with group normalization, we attain favorable scaling properties of PTR with parameters: the more the parameters, the better the performance as shown in Figure 6. We also observed that choosing an appropriate method for converting the 3-dimensional feature-map tensor produced by the ResNet into a one-dimensional embedding plays a crucial role for learning accurate Q-functions and obtaining functioning policies. Unlike standard ResNet architectures for supervised learning, simply computing global average pooling (as used in many classification architectures) performs poorly. Instead we point-wise multiply the learned feature-map with a 3-dimensional parameter tensor before computing sums over the spatial dimensions which allows the network to explicitly encode spatial information. We refer to this technique as "**learned spatial embeddings**". An illustration of this architecture is provided in Figure 2. As detailed in Appendix E, Table 11, we find that utilizing this leads to improved performance.

Next, we found that a Q-function $Q_\phi(\mathbf{s}, \mathbf{a})$ obtained by running naïve multi-task CQL tends to not use the action input $\mathbf{a}$ effectively, due to strong correlations between $\mathbf{s}$ and $\mathbf{a}$ in the offline data, which is almost always the case for close-to-optimal trajectories. As a result, policy improvement against such a Q-function overfits to these correlations, producing poor policies. To resolve this issue, we modified the architecture of Q-network, to **pass the action a as input at each fully connected**

**layer**, which (as shown in Figure 2 and Appendix E, Table 12), greatly alleviates the issue, and significantly improves the performance (significantly compared to naïve CQL).

**Cross-validation after finetuning.** Since we wish to learn task-specific policies that do not overfit to small amounts of data, thereby losing their generalization ability, we must apply the right number of gradient steps during finetuning: too few gradient steps will produce policies that do not succeed at the target tasks, while too many gradient steps will give policies that have likely lose the generalization ability of the pre-trained policy. To handle this trade-off, we adopt the following heuristic as a loose guideline: we run fine-tuning for many iterations while also plotting the learned Q-values over a held-out dataset of trajectories from the target task. Then, we pick the checkpoint for which the learned Q-values most closely resemble a valid Q-function, which must increase over the time steps in the trajectory (see Figure 3; Appendix F). Empirically we find that this heuristic guides us to identify good checkpoints (more details in Appendix F).

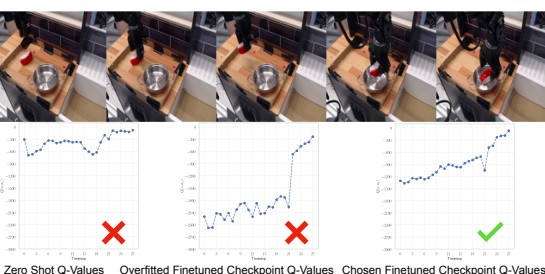

Figure 3: **Top:** PTR policy rollout of task "put sushi in pot" re-targeted to metal-pot. **Bottom: left:** Q-value over time for a target task trajectory before fine-tuning begins (zero-shot), **middle:** Q-values for a checkpoint that has started to overfit after being trained for long and exhibits drastic changes in Q-values over the course of a trajectory, and **right:** Q-values for a checkpoint that attains high performance.

**Reward specification.** In this paper, we aim to pre-train on existing robotic datasets, such as the bridge dataset (Ebert et al., 2021), which consists of human-teleoperated demonstration data. Although the demonstrations are all successful, they are not annotated with any reward function. Perhaps an obvious choice is to label the last transition of each trajectory as a success, and give it +1 binary reward. However, in several of the datasets we use, there can be a 0.5-1.0 second lag between task completion and when the episode is terminated by the data collection. To ensure that a successful transition is not incorrectly labeled as 0, we utilized the practical heuristic of annotating the last $n = 3$ transitions of every trajectory with a reward of +1 and and annotated other states with a 0 reward. We show in Appendix C that this provided the best results. In principle, more complicated methods of reward labeling (Eysenbach et al., 2021) could be used. However, we found the presented rule to be simple and yet effective to learn good policies.

## 5 EXPERIMENTAL EVALUATION OF PTR AND TAKEAWAYS FOR ROBOTIC RL

The goal of our experiments is to validate if PTR can learn effective policies from only a handful of user-provided demonstrations for a given target task, by effectively utilizing previously-collected robotic datasets for pre-training. We also aim to understand whether the design decisions introduced in Section 4.2 are crucial for attaining good robotic manipulation performance. To this end, we evaluate PTR in a variety of robotic manipulation settings, and compare it to state of the art methods that either do not use any form of offline RL or do not learn end-to-end by employing some form of visual representation learning. We are considering three scenarios: **(a)** when the target task requires retargeting the behavior of an existing skill, in this case changing the type of object types it interacts with, **(b)** when the target task requires performing a previously observed task but this time in a previously unseen domain, and **(c)** when the target task requires learning a new skill in a new domain, by using the target demonstrations. For more details and visuals, please visit our anonymous website at `https://bit.ly/PTR_ICLR`.

**Real-world experimental setup.** We directly utilize the publicly available *bridge dataset* Ebert et al. (2021) for pre-training, as it provides a large number of robot demonstrations for a diverse set of tasks in multiple domains, i.e. multiple different toykitchens. We use the same WidowX250 robot platform for our evaluations. The bridge dataset contains distinct tasks, each differing in terms of the objects that the robot interacts with and the domain the task is situated in. We assign a different task identifier to each task in the dataset for pre-training. We also evaluate on an additional door-opening task not present in the bridge dataset, where we collected demonstrations for opening and closing a variety of doors, and test our system on new, unseen doors. More details are in Appendix A.

**Comparisons.** Since the datasets we use (both the pre-training bridge dataset from (Ebert et al., 2021) and the newly collected door opening data) consist of human demonstrations, as indicated by

**Figure 4: Illustrations of the three real-world experimental setups we evaluate PTR on: (a)** the "put sushi in a metallic pot" task which requires retargeting, **(b)** the task of opening an unseen door, and **(c)** fine-tuning on several novel target tasks in a held out toykitchen environment.

prior work (Mandlekar et al., 2021), the strongest prior method in this setting is behavioral cloning (BC), which attempts to simply imitate the action of the demonstrator based on the current state. We incorporate BC in a pipeline similar to PTR, denoted as **BC (finetune)**, where we first run BC on the pre-training dataset, and then finetune it using the demonstrations on the target task using the same batch mixing as in PTR. Additionally, we tune the hyperparameters of all our BC baselines against the validation error on a held-out offline dataset. Next, to assess the importance of performing pre-training *followed* by fine-tuning, we compare PTR to **(i)** the COG approach of Singh et al. (2020), which is equivalent to jointly training with CQL on the pre-training data and the target task data from scratch, and **(ii)** multi-task offline CQL (**CQL (0-shot)**) that does not use the target demonstrations at all. We also make the analogous comparison for BC, jointly training BC on the pre-training and target task data from scratch (**BC (joint)**) which is equivalent to (Ebert et al., 2021). For fairness of comparison, BC, CQL, and PTR (both for 0-shot, joint-training and fine-tuning) use the *same* exact architecture including our learned-spatial embedding described in subsection 4.2. In Scenario 3, we compare PTR to other state-of-the-art representation learning methods, and policy architectures. More implementation details can be found on our anonymous website.

**Scenario 1: Re-targeting skills for existing tasks to act on new objects during finetuning.** We utilized the subset of the bridge data with all pick-and-place tasks in one toy kitchen for pre-training, and selected the "put sushi in pot" task as our target task. This task is demonstrated in the bridge dataset, but only using an orange transparent pot (see Figure 4 (a)). In order to pose a scenario where the offline policy at the end of pre-training must be re-targeted to act on a different object, we collected only *ten* demonstrations that place the sushi in a metallic pot. This scenario is challenging since the metallic pot drastically differs from the orange transparent pot visually. By pre-training on all pick-and-place tasks in this domain (32 tasks) and jointly fine-tuning on this data and 10 demonstrations, PTR is able to obtain a policy that is re-targeted towards the metal pot. BC appears to be mistaking arbitrary patches on the tabletop with the pot. Quantitatively, observe in Table 1, that PTR is able

| Method | Success rate |
|---|---|
| BC (0-shot) | 0/30 |
| BC (finetune) | 0/30 |
| CQL (0-shot) | 2/30 |
| **PTR (Ours)** | **14/30** |

Table 1: **Performance of PTR for "put sushi in metallic pot" in Scenario 1.** PTR substantially outperforms BC (finetune), even though it is provided access to only demonstration data. We also show some examples comparing some trajectories of BC and PTR in Appendix C.

to complete the task with reasonable accuracy, whereas 0-shot and fine-tuned BC are completely unable to solve the task. The fact that 0-shot CQL has great difficulty solving the task indicates that target demonstrations are necessary for solving this task, and PTR is able to make efficient use of these demonstrations (prior work Ebert et al. (2021) also found BC performs poorly with even 50 demonstrations).

**Scenario 2: Generalizing to previously unseen domains.** Next, we study whether PTR can adapt behaviors seen in the pre-training data to new domains. We study a door opening task, which re-

| | | | | 0-shot | | Joint Training | | Target data only | |
|---|---|---|---|---|---|---|---|---|---|
| Task | PTR (Ours) | BC (fine.) | CQL | BC | COG | BC | CQL | BC |
| Open Door | **12/20** | 10/20 | 0/20 | 0/20 | 5/20 | 7/20 | 4/20 | 7/20 |

Table 2: **Successes vs. total trials for opening a new target door in Scenario 2.** PTR outperforms both BC (finetune) and BC (joint) given access to the same data. Note that joint training is worse than finetuning from the pre-trained initialization.

quires significantly more complex maneuvers and precise control compared to the pick-and-place tasks from above (video on the anonymous website). The target door (shown in Figure 4(b)) we wish to open and the corresponding toy kitchen domain is never seen previously in the pre-training

data, and doors in the pre-training data exhibit different sizes, shapes, handle types and visual appearances. Due to the limited number of demonstrations and the associated task complexity, in order to succeed, an algorithm must effectively leverage the pre-training data. Concretely, for pre-training, we used a dataset of 800 door-opening demonstrations on 12 different doors in 4 different toy kitchen domains, and we utilize 15 demonstrations on a held-out door for finetuning. Table 2 shows that PTR improves over both BC baselines and COG.

Interestingly, Table 2 shows that while COG by itself does not outperform BC (joint), the pre-training and fine-tuning approach in PTR leads to significantly better performance, improving over the best BC baseline despite using the same CQL algorithm as COG. Since CQL (joint) is equivalent to PTR, but with no Phase 1, this large performance gap indicates the efficacy of offline RL methods trained on large diverse datasets at providing good initializations for learning new downstream tasks. We believe that this finding may be of independent interest to robotic offline RL practitioners: when utilizing multi-task offline RL, it might be better first to run multi-task pre-training followed by fine-tuning, as opposed to jointly training from scratch.

| | | BC finetuning | | | Joint training | | Target data only | | Pre-train. rep. + BC finetune | | Meta-learning |
|---|---|---|---|---|---|---|---|---|---|---|---|
| Task | PTR (Ours) | BC (fine.) | Autoreg. BC | BeT | COG | BC | CQL | BC | R3M | MAE | MACAW |
| Take croissant from metal bowl | **7/10** | 3/10 | 5/10 | 1/10 | 4/10 | 4/10 | 0/10 | 1/10 | 1/10 | 3/10 | 0/10 |
| Put sweet potato on plate | **7/20** | 1/20 | 1/20 | 0/20 | 0/20 | 0/20 | 0/20 | 0/20 | 0/20 | 1/20 | 0/20 |
| Place knife in pot | **4/10** | 2/10 | 2/10 | 0/10 | 1/10 | 3/10 | 3/10 | 0/10 | 0/10 | 0/10 | 0/10 |
| Put cucumber in pot | **5/10** | 0/10 | 1/10 | 0/10 | 2/10 | 1/10 | 0/10 | 0/10 | 0/10 | 0/10 | 0/10 |

Table 3: **Performance of PTR and other baseline methods for new tasks in Scenario 3.** Note that PTR outperforms all other baselines including BC (finetune), BC with more expressive policy classes (BeT (Shafiullah et al., 2022), Auto-regressive), representation learning methods (Nair et al., 2022; He et al., 2021) and offline RL with no pre-training ("Target data only") and joint training (Singh et al., 2020; Ebert et al., 2021).

**Scenario 3: Learning to solve new tasks in new domains.** Unlike the two scenarios studied above, in this scenario, we attempt to solve a new task in a new kitchen scene. This task is represented via a unique task identifier, and we are not provided with any data for this task identifier, or even any data with from the kitchen scene where this task is situated during pre-training. We pre-train on all 80 pick-and-place style tasks from the bridge dataset, while holding out any data from the new task kitchen scene, and then fine-tune on 10 demonstrations for 4 target tasks independently in this new kitchen, as shown in Table 3. Recent representation learning methods (R3M, MAE+BC) or more expressive policy architectures (Auto-regressive and BeT) do not lead to improved performance compared to the standard BC (finetune) approach, and we find that PTR outperforms all of these approaches. Please find more details on the implementation of R3M, MAE+BC Auto-regressive and BeT in Appendix D. This might appear surprising, and perhaps just a hyperparameter tuning artifact at first, but we present additional qualitative and quantitative analysis aiming at understanding the reasons behind why our offline RL-based PTR approach works better in the following paragraph. Figure 5 shows a comparison of rollouts of the final policy found by PTR with rollouts from a BC finetuning policy on the "take croissant from metal bowl" and "put cucumber in bowl" tasks. We also present additional rollouts in our anonymous website.

**Understanding why PTR outperforms BC baselines.** One natural question to ask given the results in this paper is: why does utilizing an offline RL method for pre-training and finetuning as in PTR outperform BC-based methods even though the dataset is quite "BC-friendly", consisting of only demonstrations? The answer to this question is not obvious, especially since joint training with BC (BC (joint)) still outperforms COG in our results in Table 3. Intuitively, we might expect that the RL-based PTR method might be better able to identify important decision points in the data, thus learning more control-

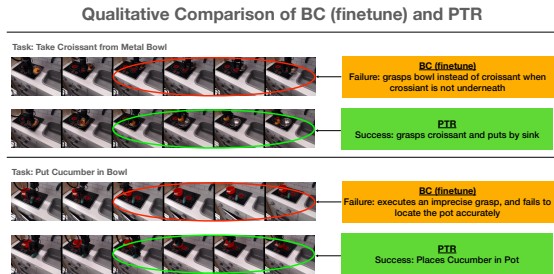

Figure 5: **Qualitative successes of PTR visualized alongside failures of BC (finetune).** As an example, observe that while PTR is accurately able to reach to the croissant and grasp it to solve the task, BC (finetune) is imprecise and grasps the bowl instead of the croissant resulting in failure.

centric representations, as suggested in prior work (Kumar et al., 2022). But can we analyze this question more precisely?

To understand the reason behind improvements from RL, we perform a qualitative evaluation of the policies learned by PTR and BC (finetune) on two tasks: take croissant from metal bowl and put cucumber in bowl in Figure 5. We find that the failure mode of BC policies can be primarily explained as a lack of precision in locating the object, or a prematurely-executed grasping action. This is especially prevalent in settings where the object of interest is farther away from the robot gripper at the initial state, and hints at the inability of BC to prioritize learning the critical decisions (e.g., precisely moving over the object before the grasping action) over non-critical ones (e.g., the action to take to reach nearby the object from farther away). On the other hand, RL can learn to make such critical decisions correctly as shown in Figure 5.

More concretely, to verify if the performance benefits can be explained entirely by the ability of Q-learning to prioritize critical decisions, we run a version of weighted behavioral cloning, where the weights are derived from the *advantage estimates computed using a frozen Q-function learned by PTR* after fine-tuning: $w_\phi(\mathbf{s}, \mathbf{a}) = \exp(Q_\phi(\mathbf{s}, \mathbf{a}) - \max_{\mathbf{a}'} Q_\phi(\mathbf{s}, \mathbf{a}'))$. As shown in Table 6, we find that this advantage-weighted BC (AW-BC) approach performs significantly better than BC (finetune) method and

| Task | BC (finetune) | PTR ‖ AW-BC (finetune) |
|---|---|---|
| Cucumber | 0/10 | 5/10 ‖ 5/10 |
| Croissant | 3/10 | 7/10 ‖ 6/10 |

Table 4: **Performance of advantage-weighted BC** on two tasks from Table 3. Observe that weighting the BC objective using advantage estimates from the Q-function learned by PTR leads to much better performance than standard BC (finetune), almost recovering PTR performance. This test indicates that the Q-function in PTR allows us to be accurate on the more critical decisions, thereby preventing the failures of BC.

comparably to PTR, for two tasks (croissant and cucumber from Table 3. Since AW-BC is essentially the same as BC, just with a modified weight to indicate the importance of any transition, this performance improvement clearly indicates the benefits of learning value functions via RL in a pre-training then fine-tuning setting, even when we only have demonstration data. Note that since AW-BC uses the PTR-weights after fine-tuning, it cannot serve as an independent method.

**Our design decisions enable us to effectively leverage high-capacity neural networks.** To understand the importance of designing techniques that enable us to use high-capacity models for offline RL, we examine the efficacy of PTR with different Q-networks, on the open door task from Scenario 2, and the put cucumber in pot and take croissant out of metallic bowl tasks from Scenario 3. We compare to standard three-layer convolutional network architectures used by prior work for DM-control tasks (see for example, Kostrikov et al. (2020)), an IMPALA (Espeholt et al., 2018) ResNet that consists of 15 convolutional layers spread across a stack of 3 residual blocks, and the ResNet 18, 34, and 50 architectures with our proposed design decisions. Observe in Figure 6, that the performance of smaller networks (Small, IMPALA) is significantly worse than the ResNet in the door opening task. For the pick-and-place tasks that contain a much larger dataset, Small, IMPALA and ResNet18 all perform much worse than ResNet 34 and ResNet

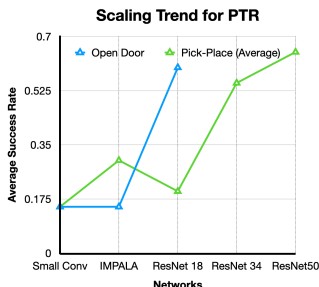

Figure 6: **Scaling trends for PTR** on the open door task from Scenario 2, and average over two pick and place tasks from Scenario 3. Note that with our design decisions, PTR is able to effectively benefit from high capacity function approximators.

50. In Appendix E we show that ResNet 34 models perform much worse if our prescribed design decisions are not used. We also perform a diagnostic study in simulation, whose details can be found in Appendix C (Table 7), and these support our real-world results.

## 6 CONCLUSION AND DISCUSSION

We presented a system that uses diverse prior data for general-purpose offline RL pretraining, followed by fine-tuning to downstream tasks. The prior data, sourced from a publicly available dataset, consists of over a hundred tasks across ten scenes and our policies can be fine-tuned with as few as 10 demonstrations. We show that this approach outperforms prior pre-training and fine-tuning methods based on imitation learning. One of the most exciting directions for future work is to further scale up this pre-training to provide a single policy initialization, that can be utilized as a starting point, similar to GPT3 (Brown et al., 2020). A limitation of our method is that it requires the prior data and new tasks to be structurally similar and an exciting future direction is to scale it up to more complex settings, including to novel robots. As we observed that joint training with offline RL (COG) was worse than pre-training and then fine-tuning with PTR, another exciting direction for future work is to understand the pros and cons of joint training and fine-tuning in the context of robot learning.

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

# Appendices

## A  DETAILS OF OUR EXPERIMENTAL SETUP

### A.1  REAL-WORLD EXPERIMENTAL SETUP

A picture of our real-world experimental setup is shown in Figure 7. The scenarios considered in our experiments (Section 5) are designed to evaluate the performance of our method under a variety of situations and therefore we set up these tasks in different toykitchen domains (see Figure 7) on three different WidowX 250 robot arms. We use data from the bridge dataset (Ebert et al., 2021) consisting of data collected with many robots in many domains for training but exclude the task / domain that we use for evaluation from the training dataset.

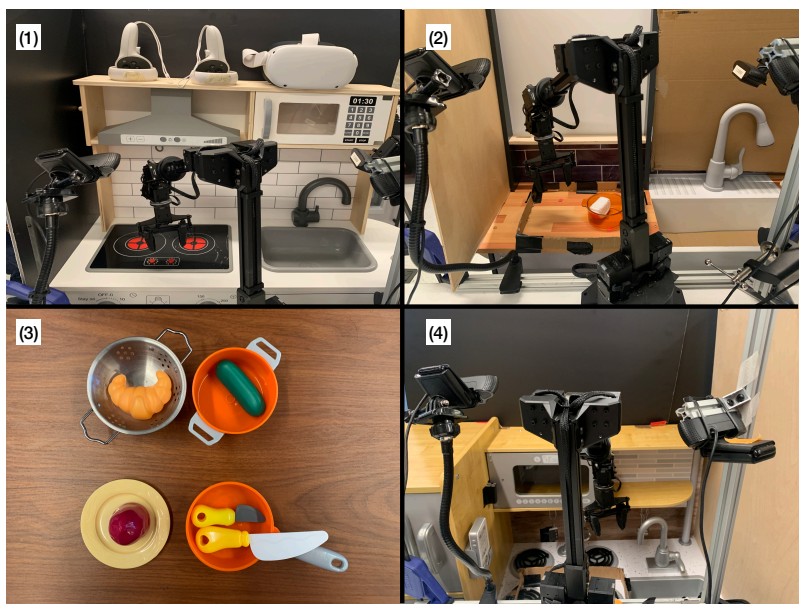

Figure 7: **Setup Overview**: Following Ebert et al. (2021), we use a toykitchen setup described in that prior work for our experiments. This utilizes a 6-DoF WidowX 250 robot. **(1):** Held-out toykitchen used for experiments in Scenario 3 (denoted "toykitchen 6"), **(2):** Re-targeting toykitchen used for experiments in Scenario 2 (denoted "toykitchen 2"), **(3):** target objects used in the experiments of scenario 3., **(4):** the held-out kitchen setup used for door opening ("toykitchen 1").

### A.2  DIAGNOSTIC EXPERIMENTAL SETUP IN SIMULATION

In simulation, we evaluate our approach in a simulated bin-sorting task on the simulated WidowX 250 platform, aimed to mimic the setup we use for our real-world evaluations. This setup is designed in the PyBullet simulation framework provided by Singh et al. (2020). A picture is shown in Figure 8. In this task, two different bins and two different objects are placed in front of the WidowX robot. The goal of the robot is to correctly sort each of the two objects to their designated bin (e.g the cylinder is supposed to be placed in the left bin and teapot should be placed in the right bin. We refer to this task as a *compound* task since it requires successfully combining behaviors of two different pick-and-place skills one after the other in a single trajectory while also adequately identifying the correct bin associated with each object. A success is counted only when the robot can accurately sort *both* of the objects into their corresponding bins.

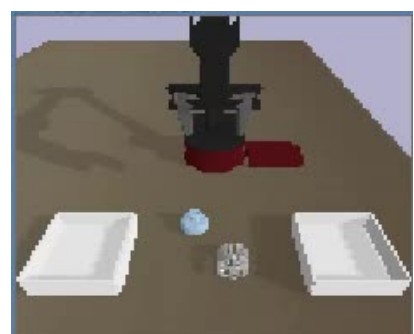

Figure 8: **Bin-Sorting task used for our simulated evaluations.** The task requires sorting the cylinder into the left bin and the teapot into the right bin.

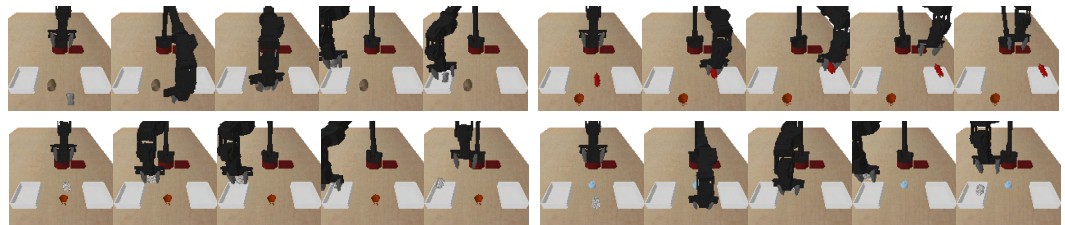

Figure 9: Some trajectories from the pre-training data used in the simulated bin-sort task.

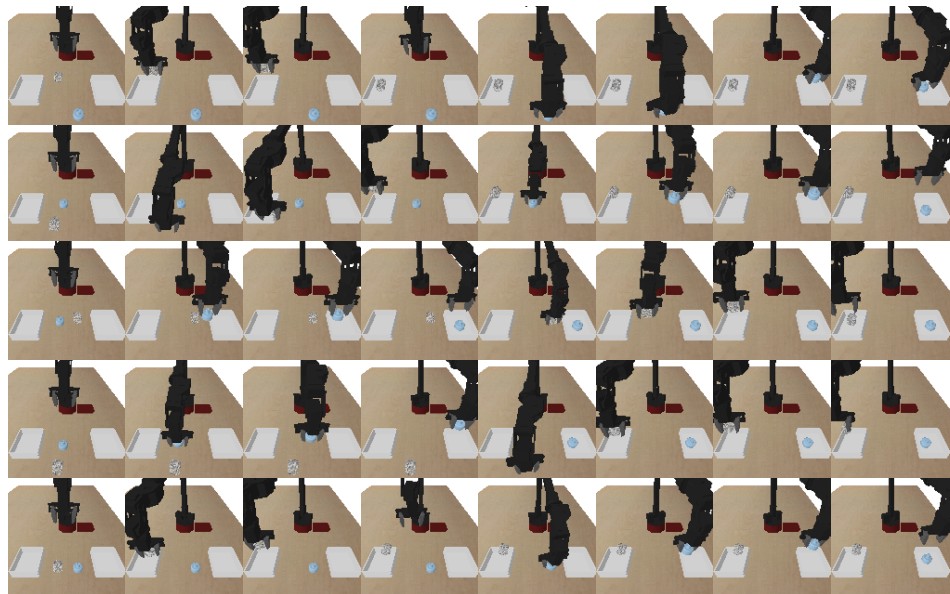

Figure 10: The five demonstration trajectories used for Phase 2 of PTR.

**Offline pre-training dataset.** The dataset provided for offline pre-training only consists of demonstrations that show how the robot should pick one of the two objects and place it into one of the two bins. Each episode in the pre-training dataset is about 30-40 timesteps long. A picture showing some trajectories from the pre-training dataset are shown in Figure 9. While the downstream task only requires solving this sorting task with two specific objects (shown in Figure 10), the pre-training data consists of a 10 unique objects (some shown in Figure 9). The two target objects that appear together in the downstream target scene are never seen together in the pre-training data. Since the pre-training data only demonstrates how the robot must pick up one of the objects and place it in one of the two bins (not necessarily in the target bin that the target task requires), it neither consists of any behavior that places objects into bins sequentially, nor does it consist of any behavior where one of objects is placed one of the bins while the other one is not. This is what makes this task particularly challenging.

**Target demonstration data.** The target task data provided to the algorithm consists of only *five* demonstrations that show how the robot must complete both the stages of placing both objects (see Figure 10). Each episode in the target demonstration data is 80 timesteps long, which is substantially longer than any trajectory in the pre-training data, though one would hope that good representations learned from the pick and place tasks are still useful for this target task. While all methods are able to generally solve the first segment of placing the first object into the correct bin, the primary challenge in this task is to effectively sort the second object, and we find that PTR attains a substantially better success rate than other baselines in this exact step.

## B    DESCRIPTION OF THE REAL-WORLD EVALUATION SCENARIOS

In this section, we describe the real-world evaluation scenarios considered in Section 5. We additionally include a much more challenging version of Scenario 3, for which we present results in Appendix C. These harder test cases evaluate the finetuning performance on four different tasks,

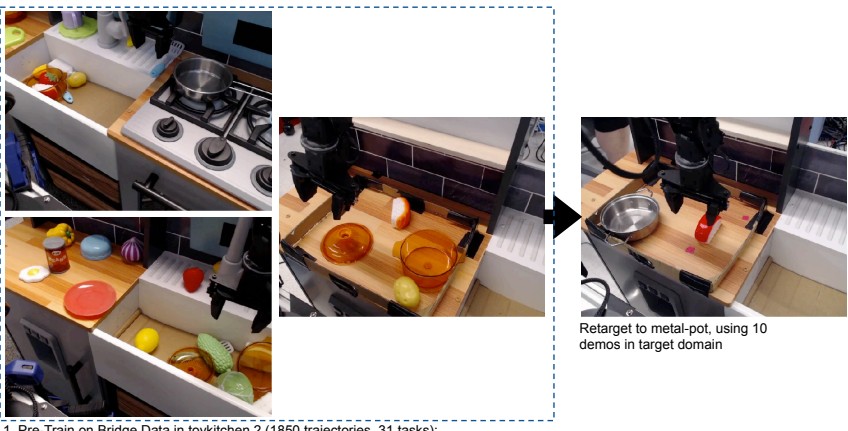

Figure 11: **Illustration of pre-training data and finetuning data used for Scenario 1**: re-targeting the put sushi in metal-pot behavior to put the object in the metal pot instead of the orange transparent pot.

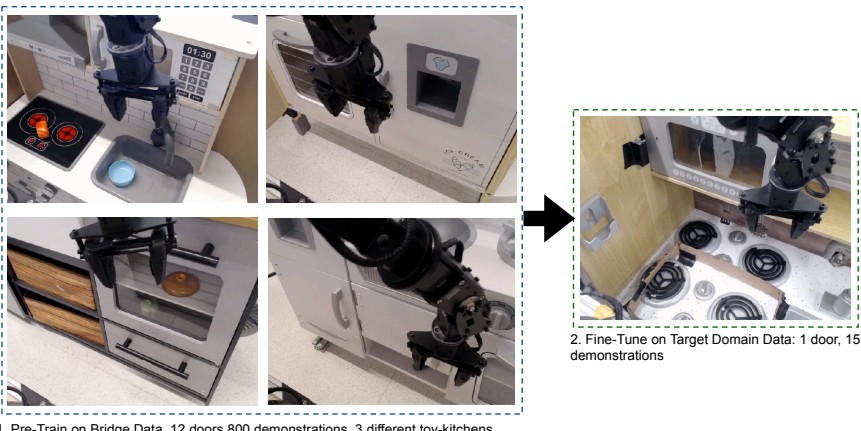

Figure 12: **Illustration of pre-training data and fine-tuning data used for Scenario 2 (door opening)**: transferring a behavior to a held-out domain.

starting from the same initialization trained on bridge data except the toykitchen 6 domain in which these four tasks were set up. In the following sections, nomenclature for the toy kitchens is drawn from Ebert et al. (2021) and as described in the caption of Figure 7.

### B.1 SCENARIO 1: RE-TARGETING SKILLS FOR EXISTING TO SOLVE NEW TASKS

**Pre-training data.** The pre-training data comprises of all of the pick and place data from the bridge dataset (Ebert et al., 2021) from toykitchen 2. This includes data corresponding to the task of putting the sushi in the transparent orange pot (Figure 11).

**Target task and data.** Since our goal in this scenario is to re-target the skill for putting the sushi in the transparent orange pot to the task of putting the sushi in the metallic pot, we utilize a dataset of 20 demonstrations that place the sushi in a metallic pot as our target task data that we fine-tune with (shown in Figure 11).

**Quantitative evaluation protocol.** For our quantitative evaluations in Table 1, we run 10 controlled evaluation rollouts that place the sushi and the metallic pot in different locations of the workspace. In all runs the arm starts at up to 10 cm distance above the target object. The initial object and arm poses and positions are matched as closely as possible for different methods.

### B.2 SCENARIO 2: GENERALIZING TO PREVIOUSLY UNSEEN DOMAINS

**Pre-training data.** The pre-training data in Scenario 2 consists of 800 door opening demonstrations on 12 different doors across 3 different toykitchen domains.

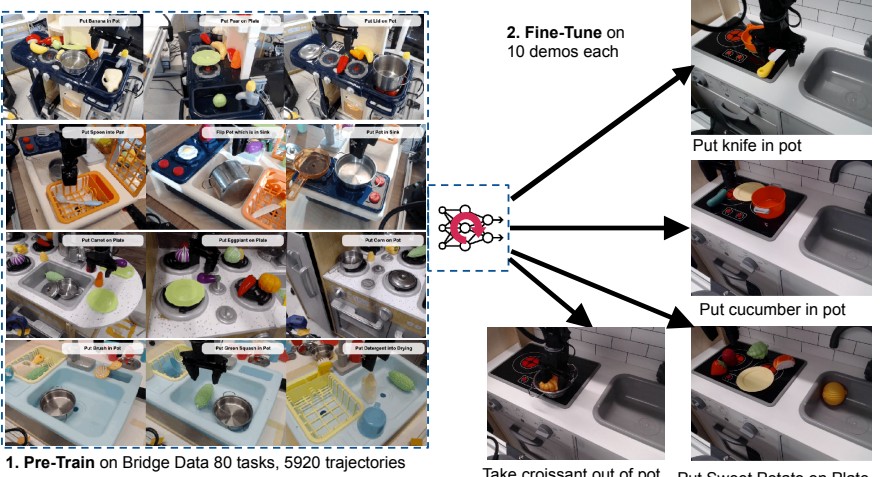

Figure 13: **Illustration of pre-training data and fine-tuning data used for the new tasks we have added in Scenario 3**. The goal is to learn to solve new tasks in new domains starting from the same pre-trained initialization and when fine-tuning is only performed using 10-20 demonstrations of the target task.

**Target task and data.** The target task requires opening the door of an unseen microwave in toykitchen 1 using a target dataset of only 15 demonstrations.

**Quantitative evaluation protocol.** We run 20 rollouts with each method, counting successes when the robot opened the door by at least 45 degrees. To perform this successfully, there is a degree of complexity as the robot has to initial open the door till it's open to about 30 degrees. Then due to physical constraints, the robot needs to wrap around the door and push it open from the inside. To begin an evaluation rollout, we reset the robot to randomly sampled poses obtained from held-out demonstrations on the target door. This is a compound task requiring the robot to first grab the door by the handle, next move around the door, and finally push the door open. As before, we match the initial pose of the robot as closely as possible for all the methods.

### B.3 Scenario 3: Learning to Solve New Tasks in New Domains

**Pre-training data.** All pick-and-place data in the bridge dataset (Ebert et al., 2021) except any demonstration data collected in toykitchen 6, where our evaluations are performed.

**Target task and data.** The target task requires placing a corn in a pot in the sink in the new target domain and the target dataset provides 10 demonstrations for this task. These target demonstrations are sampled from the bridge dataset itself.

**Quantitative evaluation protocol.** During evaluation we were unable to exactly match the camera orientation used to collect the target demonstration trajectories, and therefore ran evaluations with a slightly modified camera view. This presents an additional challenge for any method as it must now generalize to a modified camera view of the target toykitchen domain, without having ever observed this domain or this camera view during training. We sampled initial poses for our method by choosing transitions from a held out dataset of demonstrations of the target task and resetting the robot to those initial pose for each method. We attempted to match the positions of objects across methods as closely as possible.

### B.4 More Tasks in Scenario 3: Learning to Solve Multiple New Tasks in New Domains From the Same Initialization

In Appendix C, we have now added results for more tasks in Scenario 3. The details of these tasks are as follows:

**Pre-training data.** All pick-and-place data from bridge dataset (Ebert et al., 2021) except data from toykitchen 6.

**Target task and data.** We consider four downstream tasks: take croissant from a metallic bowl, put sweet potato on plate, place knife in pot, and put cucumber in bowl, and collected 10 target demon-

strations for croissant, sweet potato and put cucumber in bowl tasks, and 20 target demonstrations for the knife in pot task. A picture of these target tasks is shown in Figure 13.

**Qualitative evaluation protocol.** For our evaluations, we utilize either 10 or 20 evaluation rollouts. As with all of our other quantitative results, we evaluate all the baseline approaches and PTR starting from an identical set of initial poses for the robot. These initial poses are randomly sampled from the poses that appear in the first 10 timesteps of the held-out demonstration trajectories for this target task. For the configuration of objects, we test our policies in a variety of task-specific configurations that we discuss below:

- **Take croissant from metallic bowl:** For this task, we alternate between two kinds of positions for the metallic bowl. In the "easy" positions, the metallic bowl is placed roughly vertically beneath the robot's initial starting pose, whereas in the "hard" positions, the robot must first move itself to the right location of the bowl and then execute the policy.

- **Put cucumber in bowl:** We run 10 evaluation rollouts starting from 10 randomly sampled initial poses of the robot for our evaluations. Here we moved the bowl between the two stovetops in each trial.

- **Put sweet potato on plate:** For this task, we performed 20 evaluation rollouts. We only sampled 10 initial poses for the robot, but for each position, we evaluated every policy on two orientations of the sweet-potato (i.e., the sweet potato is placed on the table on its flat face or on its curved face). Each of these orientations present some unique challenges, and evaluating both of them allows us to gauge how robust the learned policy is to changes in orientation. The demonstration data had a variety of orientations for the sweet potato object that differed for each collected trajectory.

- **Place knife in pot:** We evaluate this task over 10 evaluation rollouts, where the first five rollouts use a smaller knife, while the other five rollouts use a larger knife (shown in Figure 7). Each knife was seen in the demanstration dataset with equal probability.

We will discuss the results obtained on these new tasks in Appendix C.

## C ADDITIONAL EXPERIMENTAL RESULTS

**Croissant Task Multiple Viewpoint Experiment**

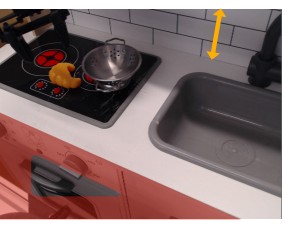 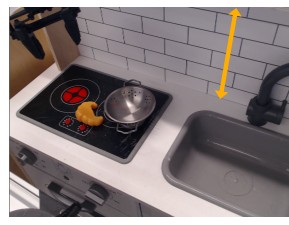 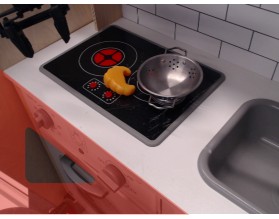

Original Viewpoint        Elevated Viewpoint        Rotated Viewpoint

Figure 14: **Sample observations from different camera viewpoints, only used during fine-tuning. Left:** the original camera viewpoint found in Figure 13. **Middle:** an elevated camera viewpoint where the robot and camera has been raised 7 cm. **Right:** a rotated camera viewpoint where the kitchen has been slightly translated and rotated 15 degrees counterclockwise relative to the camera and robot.

**Finetuning to novel camera viewpoints:** Even though Scenario 3 already presents a novel toy-kitchen domain and previously unseen objects during finetuning, we also evaluate PTR on a more challenging scenario where we additionally alter the camera viewpoint during finetuning. We apply two kinds of alterations to the camera: **(a)** we elevate the mount platform of the camera by 7 cm, which necessitates adapting the way the physical coordinates of the robot end-effector are interpreted by the policy, and **(b)** we rotate the camera by about 15 degrees to induce a more oblique image observation than what was ever seen during pre-training. Note that in both of these scenarios, the robot has never encountered such camera viewpoints during pre-training, which makes this scenario even more challenging. The original dataset in (Ebert et al., 2021) had the camera elevated to the same position for eadch domain and always ensured the kitchen was parallel to the camera platform, with translations being the primary changes in scene for each domain. In Table 5, we present our results comparing PTR and BC (finetune). Observe that PTR still clearly outperforms BC (finetune),

and attains performance close to that of PTR in Table 3, indicating that such shifts in the camera do not drastically hurt PTR.

| Method | Elevated Viewpoint | Rotated Viewpoint |
|---|---|---|
| BC (finetune) | 2/10 | 3/10 |
| **PTR (Ours)** | **6/10** | **7/10** |

Table 5: **Comparison of PTR and BC (finetune), when evaluated on novel camera viewpoints** with elevated and rotated cameras as shown in Figure 14 for the croissant task. Observe that PTR still outperforms BC (finetune) in this setting and attains more than 2x success rate of BC (finetune).

## C.1 EXPANDED DISCUSSION: WHY DOES PTR OUTPERFORM BC-BASED METHODS, EVEN WITH DEMONSTRATION DATA?

One natural question to ask given the results in this paper is: why does utilizing an offline RL method for pre-training and finetuning as in PTR outperform BC-based methods even though the dataset is quite "BC-friendly", consisting of only demonstrations? One might speculate that an answer to this question is that our BC baseline can be tuned to be much better. However, note that our BC baseline is not suboptimally tuned. We utilize the procedure prescribed by prior work (Ebert et al., 2021) for tuning BC as we discuss in Appendix D. In addition, the fact that **BC (joint)** does actually outperform **CQL (joint)** in many of our experiments, indicates that our BC baselines are well tuned. To explain the contrast to Ebert et al. (2021), note that the setup in this prior work utilized many more target task demonstrations ($\geq 50$ demonstrations from the target task) compared to our evaluations, which might explain why our BC-baseline numbers are lower in an absolute sense. Therefore, the technical question still remains: why would we expect PTR to perform better than BC? We will attempt to answer this question using some empirical evidence and visualizations. Also, we will aim to provide intuition for why our approach PTR outperforms the baseline.

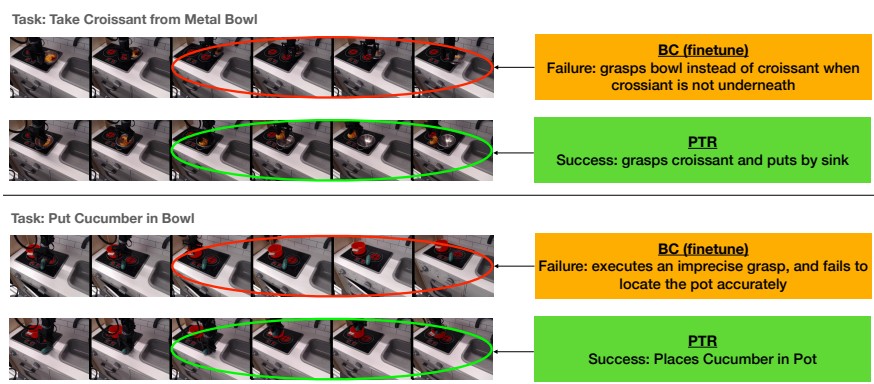

Figure 15: **Qualitative successes of PTR visualized alongside failures of BC (finetune).** As an example, observe that while PTR is accurately able to reach to the croissant and grasp it to solve the task, BC (finetune) is imprecise and grasps the bowl instead of the croissant resulting in failure.

**To begin answering this question,** it is instructive to visualize some failures for a BC-based method and qualitatively attempt to understand why BC is worse than utilizing PTR. We visualize some evaluation rollouts for **BC (finetune)** and PTR as film strips in Figure 15. Specifically, we visualize evaluation rollouts that present a challenging initial state. For example, for the rollout from the take croissant out of metallic pot task, the robot must first accurately position itself over the croissant before executing the grasping action. Similarly, for the rollout from the cucumber task, the robot must accurately locate the bowl and precisely try to grasp the cucumber. Observe in Figure 5 that **BC (finetune)** typically fails to accurately reach the objects of interest (croissant and the bowl) and executes the grasping action prematurely. On the other hand, PTR is more robust in these situations, and is able to accurately reach the object of interest before it executes the grasping action or the releasing action. Why does this happen?

**To understand why this happens**, one mental model is to appeal to the critical states argument from Kumar et al. (2022). Intuitively, this argument suggests that in tasks where the robot must precisely accomplish actions at only few specific states (called "**critical states**") to succeed, but the actions at other states (called "non-critical states") do not matter as much. Thus, offline RL-style methods can outperform BC-based methods even with demonstration data. This is because learning a value function can enable the robot to reason about which states are more important than others, and the resulting policy optimization can "focus" on taking correct actions at such critical states. Our real-world evaluation scenarios exhibit such a structure. The majority of the actions that the robot must take to reach the object do not need to be precise as long as they generally move the robot in the right direction. However, in order to succeed, the robot must critically ensure to position the arm right above the object in a correct orientation and position itself right above the container in which the object must be placed. These are the critical states and special care must be taken to execute the right action at these states. In such scenarios, the argument of Kumar et al. (2022) would suggest that offline RL should be better. We believe that we observe a similar effect in our experiments: the learned BC policies are often not precise-enough at those critical states where taking the right action is critical to succeed.

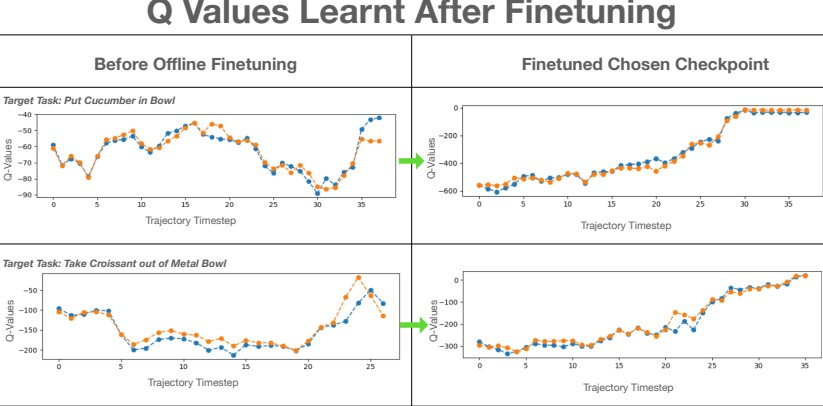

Figure 16: **Evolution of Q-values on the target task** over the process of fine-tuning with PTR. Observe that while the learned Q-values on *held-out* trajectories from the dataset just at the beginning of Phase 2 (finetuning) do not exhibit a roughly increasing trend, the checkpoint of PTR we choose to evaluate exhibits a generally increasing trend in the Q-values despite having access to only 10 demonstrations for these target tasks.

As supporting evidence to the discussion above, we further visualize the Q-values over held-out trajectories from the target demonstration data that were never seen by PTR during fine-tuning in Figure 16. To demonstrate the contrast, we present the trend in Q-values before fine-tuning and for the checkpoint selected for evaluation after fine-tuning on the target task. Observe that the Q-values for the chosen checkpoint generally increase over the course of the trajectory indicating that the learned Q-function is able to fit well to the target data. Also, the learned Q-function generalizes to held-out trajectories despite the fact that only 10 demonstrations were provided during the fine-tuning phase. This evidence supports the claim that it is reasonable to expect the learned Q-function to be able to focus on the more critical decisions in the trajectory.

**To further support our hypothesis that PTR outperforms BC-based methods because the learned value function enables us to learn about "critical" decisions**, we run an experiment that essentially runs a weighted version of BC during finetuning, where the weights are provided by exponentiated advantage values, where the advantages are defined as: $A_\theta(\mathbf{s}, \mathbf{a}) = Q_\theta(\mathbf{s}, \mathbf{a}) - \max_{\mathbf{a}'} Q_\theta(\mathbf{s}, \mathbf{a}')$ under a Q-function learned by PTR. This approaches essentially matches BC fine-tuning in all aspects: the policy parameterization, the loss function (mean-squared error), and the details of training are kept identical to our BC baselines, with the exception of an additional weight given by $\exp(A_\theta(\mathbf{s}, \mathbf{a}))$ on a given transition $(\mathbf{s}, \mathbf{a}, r, \mathbf{s}')$ observed in the set of limited task-specific demonstrations. We refer to this approach as "advantage-weighted BC finetuning".

In contrast to our BC (finetune) results from Table 3, where PTR significantly outperformed BC (finetune), observe in Table 6, that advantage-weighted BC (finetune) performs comparably to PTR on the two tasks we studied for our analysis. This result is significant since it implies that all other factors kept identical, utilizing the weights given by the Q-function is the crucial factor in improving

the performance of BC and avoids the qualitative failure modes associated with BC methods shown in Figure 15.

| Task | BC (finetune) | PTR (Ours) | Advantage-weighted BC (finetune) |
|---|---|---|---|
| Put cucumber in pot | 0/10 | 5/10 | 5/10 |
| Take croissant from metal bowl | 3/10 | 7/10 | 6/10 |

Table 6: **Performance of advantage-weighted BC** on two tasks from Table 3. Observe that weighting the BC objective using advantage-weights computed using the Q-function learned by PTR leads to much better performance than standard BC (finetune), and close to PTR. This test indicates that the Q-function in PTR allows us to focus on more critical points, thereby preventing the failures discussed in Figure 15.

## C.2 Results In Simulation

In this section, we present some additional results comparing BC-based methods and CQL to PTR in our simulated bin-sorting task. Recall that goal of this task was to solve a two-stage, compound task with only *five* target demonstrations as discussed in Figure 8, Appendix A.2. Most importantly, the pre-training data does not show any instance of the robot attempting to solve this two-stage task.

The performance numbers (along with 95%-confidence intervals) are shown in Table 7. Observe that PTR improves upon prior methods in a statistically significant manner, outperforming the BC baselines by a significant margin. This validates the efficacy of PTR in simulation, and corroborates our real-world results.

| Method | Success rate |
|---|---|
| BC (joint training) | $7.00 \pm 0.00$ % |
| COG (joint training) | $8.00 \pm 1.00$ % |
| BC (finetune) | $4.88 \pm 4.07$ % |
| **PTR (Ours)** | **$17.41 \pm 1.77$ %** |

Table 7: **Performance of PTR in comparison with other methods** on the simulated bin sorting task, trained for many more gradient steps for all methods until each one of them converges. Observe that PTR substantially outperforms other prior methods, including joint training on the same data with BC or CQL. Training on target data only is unable to recover a non-zero performance, and so we do not report it in this table. Since the 95%-confidence intervals do not overlap between PTR and other methods, it indicates that PTR improves upon baselines in a statistically significant manner.

## D Hyperparameters for PTR and Baseline Methods

In this section, we will present the hyperparameters we use in our experiments and explain how we tune the other hyperparameters for both our method PTR and the baselines we consider.

**PTR.** Since PTR utilizes CQL as the base offline RL method, it trains two Q-functions and a separate policy, and maintains a delayed copy of the two Q-functions, commonly referred to as target Q-functions. We utilize completely independent networks to represent each of these five models (2 Q-functions, 2 target Q-functions and the policy). We also do not share the convolutional encoders among them. As discussed in the main text, we rescaled the action space to $[-1, 1]^{|\mathcal{A}|}$ to match the one used by actor-critic algorithms, and utilized a tanh squashing function at the end of the policy. We used a CQL $\alpha$ value of 10.0 for our pick and place experiments. The rest of the hyperparameters for training the Q-function, the target network updates and the policy are taken from the standard training for image-based CQL from Singh et al. (2020), and are presented in Table 9 below for completeness. The hyperparameters we choose are essentially the network design decisions of **(1)** utilizing group normalization instead of batch normalization, **(2)** utilizing learned spatial embeddings instead of standard mean pooling, **(3)** passing in actions at each of the fully connected layers of the Q-network and the hyperparameter $\alpha$ in CQL that must be adjusted since our data consists of demonstrations. We will ablate the new design decisions explicitly in Appendix E.

The only other hyperparameter used by PTR is the mixing ratio $\tau$ that determines the proportion of samples drawn from the pre-training dataset and the target dataset during the offline finetuning

| Hyperparameter | Value |
|---|---|
| Q-function learning rate | 3e-4 |
| Policy learning rate | 1e-4 |
| Target update rate | 0.005 (soft update with Polyak averaging) |
| Optimizer type | Adam |
| Discount factor $\gamma$ | 0.96 (since trajectories have a length of only about 30-40) |
| Use terminals | True |
| Reward shift and scale | shift = -1, scale = 10.0 |
| CQL $\alpha$ | 10.0 |
| Use Color Jitter | True |
| Use Random Cropping | True |

Table 8: **Main hyperparameters for CQL training in our real-world experiments.** In simulation, we utilize a smaller $\alpha$ for CQL, $\alpha = 1.0$ and a larger discount $\gamma = 0.98$ since trajectories in simulation are about 60-70 timesteps in length.

phase in PTR. We utilize $\tau = 0.7$ for our experiments with PTR in the main paper, and use $\tau = 0.9$ for the additional experiments we added in the Appendix. This is because $\tau = 0.9$ (more bridge data, and smaller amount of target data) was helpful in scenarios with very limited target data.

In order to perform checkpoint selection for PTR, we utilized the trends in the learned Q-values over a set of held-out trajectories on the target data as discussed in Section 4.2. We did not tune any other algorithmic hyperparameters for CQL, as these were taken directly from (Singh et al., 2020).

**BC (finetune).** We trained BC in a similar manner as Ebert et al. (2021), utilizing the design decisions that this prior work found optimal for their experiments. The policy for BC utilizes the very same ResNet 34 backbone as our RL policy since a backbone based on ResNet 34 was found to be quite effective in Ebert et al. (2021). Following the recommendations of Ebert et al. (2021) and based on result trends from our own preliminary experiments, we chose to not utilize the tanh squashing function at the end of the policy for any BC-based method, but trained a deterministic BC policy that was trained to regress to the action in the demonstration with a mean-squared error (MSE) objective.

| Hyperparameter | Value |
|---|---|
| Policy learning rate | 1e-4 |
| Optimizer type | Adam |
| Use Color Jitter | True |
| Use Random Cropping | True |
| Dropout | 0.4 |

Table 9: **Main hyperparameters for Behavior Cloning Baseline Training in our real-world and simulation experiments.** Note: architecture design choices follow closely to PTR design choices.

In order to perform cross-validation, checkpoint and model selection for our BC policies, we follow guidelines from prior work (Ebert et al., 2021; Emmons et al., 2021) and track the MSE on a held-out validation dataset similar to standard supervised learning. We found that a ResNet 34 BC policy attained the smallest validation MSEs in general, and for our evaluations, we utilized a checkpoint of a ResNet 34 BC policy that attained the smallest MSE.

Analogous to the case of PTR discussed above, we also ablated the performance of BC for a set of varying values of the mixing ratio $\tau$, but found that a large value of $\tau = 0.9$ was the most effective for BC, and hence utilized $\tau = 0.9$ for BC (finetune) and BC (joint).

**BC (joint) and CQL (joint).** The primary distinction between training **BC (joint)** and **BC (finetune)** and correspondingly, **CQL (joint)** and PTR was that in the case of joint training, the target dataset was introduced right at the beginning of Phase 1 (pre-training phase), and we mixed the target data with the pre-training data using the same value of the mixing ratio $\tau$ used in for our fine-tuning experiments to ensure a fair comparison.

**Few-shot offline meta-RL (MACAW) (Mitchell et al., 2020):** We compare to two variants of this algorithm and perform an **extensive** sweep over several hyperparameters, shown in Table 10.

We trained two different variants of MACAW in our evaluation: **(1)** Pre-training on the bridge data in Scenario 3 and then fine-tuning on target data of interest, and **(2)** adapting a set of existing task identifiers to the target task of interest utilizing the same pre-training and fine-tuning domains. We performed early stopping on the meta-training based on validation losses. From there, we started the meta-testing phase, adapting to the target domain of interest. Following Mitchell et al. (2020), we use a task mini-batch of 8 tasks at each step of optimization rather than using all of the training tasks. We clipped the advantage weight logits to the scale of 20 and attempted to utilize a policy network with a fixed and learned standard deviation. Additionally, we varied the number of Adaptation steps following prior work. Our evaluation protocol for MACAW entails utilizing the validation losses to choose an initial checkpoint for evaluation. Then, we consider checkpoints in the neighborhood ($\pm$ 50K gradient steps) to for evaluations as well and chose the max over all of these checkpoints as the final evaluation success rate.

Quantitatively, as seen in Table 3, MACAW was unable to get a non-zero success rates on any of the tasks we study. However, we did qualitatively observe nontrivial behavior seen in our evaluation rollouts. For instance, we found that the policies trained via MACAW could consistently grasp the object of interest but were unable to localize where to place the object correctly. Several trials involved hovering around with the object of interest and not placing the object in the container. Other trials involved the agent failing to grasp the object.

| Hyperparameter | Value |
|---|---|
| Optimizer | Adam |
| Outer Policy learning rate | 1e-4 |
| Outer Value learning rate | 1e-5, 1e-6 |
| Inner Policy learning rate | 1e-2, 1e-3 |
| Inner Value learning rate | 1e-3, 1e-4 |
| Auxilary Advantage Coefficient | 1e-2, 1e-3, 1e-4 |
| Policy Parameterization | Fixed std, Learned std |
| AWR Policy Temperature | 1, 10, 20 |
| Number of Adaptation Steps | 1, 2, 3 |
| Task Batch Size | 8 |
| Train Adaptation Batch Size | 64 |
| Eval Adaptation Batch Size | 64 |
| Max Advantage Clip | 20 |
| Use Color Jitter | True |
| Use Random Cropping | True |

Table 10: **Main hyperparameters for Training MACAW (Mitchell et al., 2020) in our real-world experiments.** Note: architecture design choices follow closely to PTR design choices but hyperparameter design choices follow closely the suggestions in Mitchell et al. (2020).

**Pre-trained R3M initialization (Nair et al., 2022):** Next we compare PTR to utilizing an off-the-shelf pre-trained representation given by R3M (Nair et al., 2022). We compare to two baselines that attempt to train a MLP policy on top of the R3M state representation by using BC (finetuning) and CQL (finetuning) respectively. To ensure that this baseline is well-tuned, we tried a variety of network sizes with 2, 3 or 4 MLP layers and also tuned the hidden dimension sizes in [256, 512, 1024]. We also utilized dropout as regularization to prevent overfitting and tuned a variety of values of dropout probability in [0, 0.1, 0.2, 0.4, 0.6, 0.8]. We observe in Table 3, that on the four tasks we evaluate on, PTR outperforms R3M, which indicates that training on the bridge dataset can indeed give rise to effective visual representations that are more suited to finetuning in our setting. The numbers we report in the table are the best over each parametric policy corresponding to each hyperparameter in our abalation. Checkpoint selection was done utilizing early stopping which is the last iteration where the validation error stops decreasing. Learning curves for this baseline can be found in our Anonymous Website.

**Pre-trained MAE initialization (He et al., 2021):** We took a similar training procedure to R3M for our MAE representation. We used an MAE trained on the every image from the bridge dataset Ebert et al. (2021). We then finetuned on a specific target task with a similar ablation on network size, hidden dimension size and regularization techniques such as dropout. We observe in Table **??**,

that on the four tasks we evaluate on, PTR outperforms R3M, which indicates that training on the bridge dataset can indeed give rise to effective visual representations that are more suited to finetuning in our setting. The numbers we report in the table are the best over each parametric policy corresponding to each hyperparameter in our abalation. Checkpoint selection was done utilizing early stopping which is the last iteration where the validation error stops decreasing. Learning curves for this baseline can be found in our Anonymous Website.

**Policy expressiveness study.** We considered two policy expressiveness choices for BC to compare with our reference BC implementation that is implementated with a set of MLP layers. These first of the two choices was an **autoregressive policy** where the 7 dimensional action space was discretized into 100 bins. Each action was then predicted autoregressively conditioned on the observation, task id and the action component from the previous dimension(s). The sencond approach was with the BeT Architecture from Shafiullah et al. (2022). We utilized the reference implementation from the paper with the default suggested hyperparameters for this set of abalations. The window size for the mingpt transformer was abalated over between 1, 2, and 10.

# E    VALIDATING THE DESIGN CHOICES FROM SECTION 4.2 VIA ABLATIONS

In this section, we will present ablation studies aimed to validate the design choices utilized by PTR. We found these design choices quite crucial for attaining good performance. The concrete research questions we wish to answer are: **(1)** How important is utilizing a large network for attaining good performance with PTR, and how does the performance of PTR scale with the size of the Q-function?, **(2)** How effective is a learned spatial embedding compared to other approaches for aggregating spatial information? **(3)** Is concatenating actions at each fully-connected layer of the Q-function crucial for good performance?, **(4)** Is group normalization a good alternative to batch normalization? and **(5)** How does our choice of creating binary rewards for training affect the performance of PTR?. We will answer these questions next.

**Highly expressive Q-networks are essential for good performance.** To assess the importance of highly expressive Q-functions, we evaluate the performance of PTR with varying sizes and architectures on three tasks: the open door task from Scenario 2, and the put cucumber in pot and take croissant out of metallic bowl tasks from Scenario 3. Our choice of architectures is as follows: **(a)** a standard three-layer convolutional network typically used by prior work for DM-control tasks (see for example, Kostrikov et al. (2021a)), **(b)** an IMPALA (Espeholt et al., 2018) ResNet that consists of 15 convolutional layers spread across a stack of 3 residual blocks, **(c)** ResNet 18 with group normalization and learned spatial embeddings, **(d)** ResNet 34 that we use in our experiments, and **(e)** an even bigger ResNet 50 with group normalization and learned spatial embeddings.

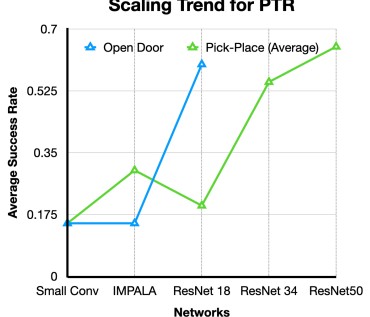

Figure 17: **Scaling trends for PTR** on the open door task from Scenario 2, and average over two pick and place tasks (take croissant out of metallic pot and put cucumber in bowl) from Scenario 3. Note that more high capacity and expressive function approximators lead to the best results.

We present our results in Figure 6. To obtain more accurate scaling trends, we plot the trend in the average success rates for the pick and place tasks from Scenario 3 along with the trend in the success rate for the open door task separately since these tasks use different pre-training datasets. Observe that the performance of smaller networks (Small, IMPALA) is significantly worse than the ResNet in the door opening task. For the pick and place tasks that contain a much larger dataset, Small, IMPALA and ResNet18 all perform much worse than ResNet 34 and ResNet 50. We believe this result is quite exciting since it highlights the possiblity of actually benefitting from using highly-expressive neural network models with TD-learning based RL methods trained on lots of diverse multi-task data (contrary to prior work (Lee et al., 2022b)). We believe that this result is a valuable starting point for further scaling and innovation.

**Learned spatial embeddings are crucial for performance.** Next we study the impact of utilizing the learned spatial embeddings for encoding spatial information when converting the feature maps from the convolutional stack into a vector that is fed into the fully-connected part of the Q-function. We compare our choice to utilizing a spatial softmax as in Ebert et al. (2021), and also global average

pooling (GAP) that simply averages over the spatial information, typically utilized in supervised learning with ResNets.

| Method | Success rate |
|---|---|
| PTR with spatial softmax | 4/10 |
| PTR with global average pooling | 4/10 |
| PTR with learned spatial embeddings **(Ours)** | **7/10** |

Table 11: **Ablation of PTR with spatial softmax and GAP on the croissant task.** Observe that PTR with learned spatial embeddings performs significantly better than using a spatial softmax or global average pooling.

As shown in Table 11 learned spatial embeddings outperform both of these prior approaches on the put croissant in pot task. We suspect that spatial softmax does not perform much better than the GAP approach since the softmax operation can easily get saturated when running gradient descent to fit value targets that are not centered in some range, which would effectively hinder its expressivity. This indicates that the approach of retaining spatial information like in PTR is required for attaining good performance.

**Concatenating actions at each layer is crucial for performance.** Next, we run PTR without passing in actions at each fully connected layer of the Q-function on the take croissant out of metallic bowl task and only directly concatenate the actions with the output of the convolutional layers before passing it into the fully-connected component of the network. On the croissant task, we find that not passing in actions at each layer only succeeds in **2/10** evaluation rollouts, which is significantly worse than the default PTR which passes in actions at each layer and succeeds in **7/10** evaluation rollouts (Table 12).

| Method | Success rate |
|---|---|
| PTR without actions passed in at each FC layer | 2/10 |
| PTR with actions passed in at each FC layer (Ours) | **7/10** |

Table 12: **Ablation of PTR with actions passed in at each layer.** Observe that passing in actions at each fully-connected layer does lead to quite good performance.

**Group normalization is more consistent than batch normalization.** Next, we ablate the usage of group normalization over batch normalization in the ResNet 34 Q-functions that PTR uses. We found that batch normalization was generally harder to train to attain Q-function plots that exhibit a roughly increasing trend over the course of a trajectory. That said, on some tasks such as the croissant in pot task, we did get a reasonable Q-function, and found that batch normalization can perform well. On the other hand, on the put cucumber in pot task, we found that batch normalization was really ineffective. These results are shown in Table 13, and they demonstrate that batch normalization may not be as consistent and reliable with PTR as group normalization.

| Method | Croissant out of metallic bowl | Cucumber in pot |
|---|---|---|
| PTR with batch norm. (relative) | + 28.0% (7/10 → 9/10) | - 60.0% (5/10 → 2/10) |

Table 13: **Relative performance of PTR with batch normalization with respect to PTR with group normalization.** Observe that while utilizing batch normalization in PTR can be sometimes more effective than using group normalization (e.g., take croissant out of metallic bowl task), it may also be highly ineffective and can reduce success rates significantly in other tasks. The performance numbers to the left of the → corresponds to the performance of PTR with group normalization and the performance to the right of → is the performance with batch normalization.

**Choice of reward function.** Finally, we present some results that ablate the choice of the reward function utilized for training PTR from data that entirely consists of demonstrations. In our main set of experiments, we labeled the last three timesteps of every trajectory with a reward of +1 and annotated all other timesteps with a 0 reward. We tried perhaps the most natural choice of labeling only the last timestep with a 0 reward on the croissant task, and found that this choice succeeds **0/10** times, compared to annotating the last three timesteps with a +1 reward which succeeds **7/10** times.

We suspect that this is because only annotating the last timestep with a +1 reward is not ideal for two reasons: first, the task is often completed in the dataset much earlier than the observation shows the task complete, and hence the last-step annotation procedure induces a non-Markovian reward function, and second, only labeling the last step with a +1 leads to overly conservative Q-functions when used with PTR, which may not lead to good policies.

## F  MORE DETAILS AND EXAMPLES OF OUR EARLY STOPPING CRITERION

Our guideline for selecting checkpoints for evaluation was to pick the checkpoints that presented a Q-function with the Q-values appearing closest to having a monotonically increasing trend in a trajectory. This is a *relative* guideline and must be performed within the checkpoints observed in a run, just like how the absolute value of validation error may not be as informative in supervised learning, as a comparison of its value across a few models. The reason for this heuristic choice is that a valid Q-function, in the end, must be a valid estimator for long-term discounted return and hence, it must increase over time-steps of demonstration trajectories for a given task. Of course, this heuristic does not hold for arbitrary sub-optimal offline data, but all of our data comes from human-collected demonstrations. In principle, this heuristic can be wrapped into a metric quantifying degree of monotonicity, but in our experiments we felt this was not necessary – as we will show below, we were able to narrow down the checkpoints to essentially one or at most, two checkpoints by just visual inspection. Of course, designing an accurate metric would be helpful for future work.

We present two worked-out examples of our checkpoint selection strategy for two tasks from Scenario 1 and Scenario 3 below. Observe that checkpoints early in training exhibit Q-values that fluctuate arbitrarily in the beginning of training, that are clearly non-monotonic. This is because the the lack of sufficient gradient steps for fine-tuning on the target task. Once sufficient gradient steps are performed, the Q-values visibly improve on the monotonicity property. Training further leads to much flatter Q-values, which start becoming more noisy and are visibly less monotonic.

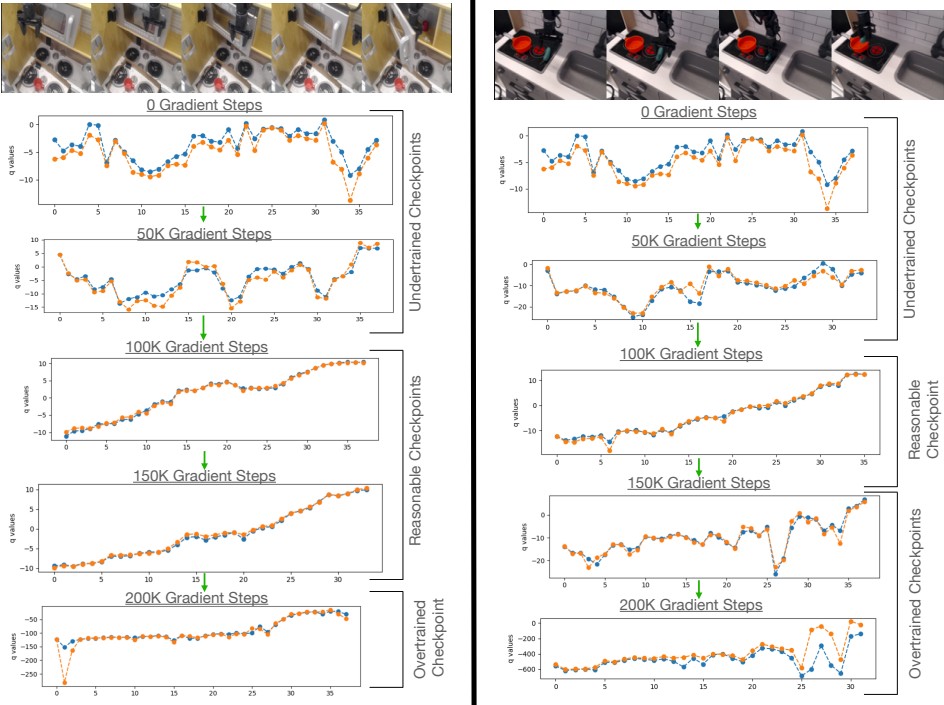

Figure 18: **Evolution of Q-values on the target task over the process of fine-tuning with PTR.** Observe that while the learned Q-values on *held-out* trajectories from the dataset just at the beginning of Phase 2 (finetuning) do not exhibit a roughly increasing trend, the checkpoint of PTR we choose to evaluate exhibits a visible more increasing trend in the Q-values despite having access to only 10 demonstrations for these target tasks.

