# OpenReview forum: "Pre-Training for Robots: Leveraging Diverse Multitask Data via Offline Reinforcement Learning"
_ICLR.cc/2023/Conference — Submitted to ICLR 2023_

### Official Review · Reviewer_WMAS · 2022-10-19

**Confidence:** 4
**Correctness:** 4
**Technical Novelty And Significance:** 2
**Empirical Novelty And Significance:** 2
**Recommendation:** 5

**Clarity, Quality, Novelty And Reproducibility:**

### **Quality**
The quality of this paper is high for proposing an empirical framework for offline RL.

### **Clarity**
The presentation is very clear.

### **Novelty**
The novelty could be improved by comparing it with existing meta-RL methods.

### **Reproducibility**
Code is not released but the technics used in the method are either well-known or easy to implement. So I think it is easy to reproduce the results.



**Strength And Weaknesses:**

### **Strength**

* Generalization of offline reinforcement learning algorithms is an interesting topic. Although there is no new algorithm or framework in this paper, the exploration of useful combinations of existing technics still contributes to this area.

* The presentation of this paper is good. All details are clearly explained.


### **Weaknesses**

* Missing baselines of offline meta-reinforcement learning [1][2]. Without comparing with existing methods, it is hard to position the empirical value of this paper.

* Sort of limited novelty. As the authors mentioned, “Although the individual components that constitute PTR are not especially innovative and are based closely on prior work, the combination of these components is novel.”Since there are already some existing novel meta-RL methods, properly comparing them could empirically improve the novelty of this paper.

---

[1] Mitchell, Eric, Rafael Rafailov, Xue Bin Peng, Sergey Levine, and Chelsea Finn. "Offline meta-reinforcement learning with advantage weighting." In International Conference on Machine Learning, pp. 7780-7791. PMLR, 2021.

[2] Xu, Mengdi, Yikang Shen, Shun Zhang, Yuchen Lu, Ding Zhao, Joshua Tenenbaum, and Chuang Gan. "Prompting decision transformer for few-shot policy generalization." In International Conference on Machine Learning, pp. 24631-24645. PMLR, 2022.



**Summary Of The Paper:**

This paper proposes a framework pre-training for robots (PTR) based on CQL that attempts to effectively learn new tasks by combining pre-training on existing robotic datasets with rapid fine-tuning on a new task, with as few as 10 demonstrations.
In particular, their method modifies CQL to include several crucial design choices that enable PTR to have strong generalization. These choices include using task indicators for policy and Q-function networks, using group normalization, and using mixed data of prior data and the target dataset.
In their experiments, they train the model on the Bridge dataset and finetune it with real-world demonstrations. Their method shows signiﬁcant improvement over prior RL-based pretraining and ﬁnetuning methods.


**Summary Of The Review:**

I acknowledge the empirical contribution of this paper to the generalization problem of offline RL. However, without a fair comparison with state-of-the-art meta-RL algorithms, I suggest rejecting this paper for now.

---

> ### Author Response · Authors · 2022-11-15
> **Author Response (Part 1 of 1): Offline Meta RL Baseline and Novelty Concerns**
>
> Thank you for your detailed feedback. To address the concern about comparisons with offline meta-RL methods, we just added a comparison with MACAW (Mitchell et al. ICML 2021) in **Table 3**, an offline meta-RL algorithm in Table 3. We observe that PTR still outperforms MACAW. MACAW attains a success rate of 0% on all tasks in scenario 3. PTR outperforms this approach on each task, averaging a success rate of 48.75%. Qualitatively, we were able to see non-naive behavior such as consistently reaching for the object of interest and lifting it. However, the approach qualitatively was having difficulty with the localization of the target container, thus leading to poor performance. We tuned the hyperparameters for MACAW extensively (see Appendix D for details of tuning; in $\textcolor{magenta}{magenta}$), but were not able to get a nonzero success rate.
>
> ___
>
> **Comparisons**
>
> We will focus on getting the comparison to the other method Xu et al. 2022 for the final version of this paper, as it requires implementing, testing, and tuning decision transformers in our implementation which will not finish in time for the rebuttal. But we have cited this paper in our related work section now.
>
> ___
>
> **With regards to novelty**, we would first emphasize that our work is primarily an application paper, and the main novel contribution is the application of offline RL to pretrain policies for a real-world end-to-end image-based robotic manipulation task. We believe that learning tasks with as few as 10 demonstrations after offline RL pretraining is a substantial novel empirical contribution, and the paper would be of interest to the community because it describes how this novel empirical result can be attained. We agree that the architectural choices are not novel, but we believe that their use in offline RL is. We believe that this is significant in itself, as no prior work has demonstrated an offline RL approach that can work well for pre-training on *existing*, open-source diverse data, that consists of entirely demonstrations. In fact, prior work (Mandlekar et al. CoRL 2021) only argues that offline RL shouldn’t be useful when provided with human demonstrations, but we show *for the first time* that offline RL can have benefits compared to imitation.
>
> We believe that the significance of this work is primarily the result that offline RL can be a viable choice for learning from diverse pre-training data consisting of demonstrations, which we believe would be valuable for the robot learning practitioners using offline RL to build upon.
>
> _____
>
> **We would appreciate it if you are willing to reconsider your rating in light of the clarifications.**

---

> ### Author Response · Authors · 2022-11-17
> **Request for discussion**
>
> Dear Reviewer,
>
> We were wondering if you have gotten the chance to go over our paper updates, and our responses to see if your concerns are resolved. We would be more than happy to resolve any remaining questions you have, and would appreciate it if you engage in a discussion with us.
>
> Thanks!

---

> > ### Author Response · Authors · 2022-11-18
> > **Follow-up**
> >
> > Dear Reviewer,
> >
> > Only a few hours are left in the discussion. We would really appreciate it if you could tell us if your concerns are resolved. We would be more than happy to resolve any remaining questions in the time we have, and would appreciate it if you engage in a discussion with us.

---

### Official Review · Reviewer_D87g · 2022-10-23

**Confidence:** 4
**Correctness:** 3
**Technical Novelty And Significance:** 2
**Empirical Novelty And Significance:** 2
**Recommendation:** 6

**Clarity, Quality, Novelty And Reproducibility:**

The paper writing is clear, except it tends to exaggerate the claims or does not define the scope of the claims properly.

**Strength And Weaknesses:**

**Strength**:

* The paper presents a few techniques, such as network architecture design and normalization trick, that are empirically shown to improve downstream task learning.

* The paper shows real-world robot experiments. This is a big plus as such experiments do require a significant amount of effort.

 * The paper compares the proposed pipeline with a decent number of baselines, including behavior cloning and especially prior works on first doing representation learning and then doing policy learning such as R3M.

**Weaknesses**:

* As the paper also says itself, the techniques presented in the paper are not new.

* Since the number of real-world tests is small (like 20?), I am not sure how one can report success rates with so many significant figures. Similarly, the paper tends to overclaim the method's effectiveness. For example, in page 5, the paper says that using "learned spatial embeddings" leads to `2x` improvement. However, if we actually look at table 10, we see that one case has 4/10 success rate and the other one has 7/10 success rate. With such a small number of tests, it is very confusing and not rigorous at all to make such claims (`2x` improvements). There are other similar claims in the paper. The authors should remove such claims.

* In table 2, I cannot find the 95% confidence interval.

* It's unclear how convincing the claim about generalizing to previously unseen domain is. From the video, I can only tell that the testing door is not that different from the training doors. Can authors add a figure showing all the training doors and the testing door? Since the claim is about generalization, it is important for readers to understand how much difference the training and testing distribution has. As a research paper, it is important to make each claim precise and accurate. While I enjoy reading the papers, especially seeing the real-world testings, reading such exaggerated claims only negatively affects my judgment.

**Summary Of The Paper:**

The paper experimented with using offline RL to pretrain and fine-tune policy and Q function. The results show that using offline RL to first pretrain the policy and Q function on a diverse offline dataset can improve the policy adaptation speed on new tasks. The paper discussed about several important design choices, such as network architectures to achieve better performance. Overall, the paper is more like an engineered work that combines some ideas from prior works and achieves fast policy learning on new tasks.

**Summary Of The Review:**

Overall I think the paper presents a decent amount of experiments and compare its proposed pipeline with some reasonable baselines. However, the paper writing needs to be improved to make the descriptions accurate.

---

> ### Author Response · Authors · 2022-11-15
> **Author Response (Part 1 of 1): Claim Clarifications, Concerns about Novelty and Number of Domains**
>
> Thank you for your detailed feedback! We have updated the paper to make the claims more precise and make clarifications on other aspects of the paper below.
>
> **We would appreciate it if you can have a look and let us know if your concerns are addressed. We are happy to answer any more concerns.**
>
> ____
>
> **I am not sure how one can report success rates with so many significant figures.** Thanks for pointing this out. The confidence intervals were computed incorrectly. We have removed those now and instead report the results in the form of the number of successes per 10 or 20 trials.
>
> **Removing strong claims** We have updated the paper to remove any strong claims, but instead directly quote the results from the Tables in the Appendix. We realize that these claims are made with 10 or 20 trials, but also note that it is actually impossible to do many more trials in real robotic setups as they take significant manual effort. We have deleted the statement mentioning 2x improvement from spatial learned embeddings.
>
> ___
>
> **It's unclear how convincing the claim about generalizing to a previously unseen domain is.  Can authors add a figure showing all the training doors and the testing door?**
>
> Here is an anonymous link to a figure showing all training doors and the testing door: https://imgur.com/DDp6Dfu . Note the variety in the sizes of the door, the orientation of the handle as well as the location of the hinge. We have now updated the anonymous website for this paper to add a complete list of images for all the training doors on our website, as well as images of the complete set of training and testing objects and environments for all other experiments (some of these are shown in Figure 7).
>
> To further demonstrate the degree of generalization to new domains, we already had an experiment in **Appendix C.1** where the fine-tuning task must be tackled from a modified camera viewpoint (from elevated and rotated camera viewpoints), even though none of the pre-training data contains this viewpoint. Observe in **Table 5** that PTR outperforms the BC (finetune) in both scenarios.
>
> ___
>
> **Novelty** We would first emphasize that our work is primarily an application paper, and the main novel contribution is the application of offline RL to pretrain policies for a real-world end-to-end image-based robotic manipulation task. We believe that learning tasks with as few as 10 demonstrations after offline RL pretraining is a substantial novel empirical contribution, and the paper would be of interest to the community because it describes how this novel empirical result can be attained. We agree that the architectural choices are not novel, but we believe that their use in offline RL is. We believe that this is significant in itself, as no prior work has demonstrated an offline RL approach that can work well for pre-training on *existing*, open-source diverse data, that consists of entirely demonstrations. In fact, prior work (Mandlekar et al. CoRL 2021) only argues that offline RL shouldn’t be useful when provided with human demonstrations, but we show *for the first time* that offline RL can have benefits compared to imitation.
>
> We believe that the significance of this work is primarily the result that offline RL can be a viable choice for learning from diverse pre-training data consisting of demonstrations, which we believe would be valuable for the robot learning practitioners using offline RL to build upon.
>
> ___
>
> **We would appreciate it if you were willing to reconsider your rating in light of the clarifications.**

---

> ### Author Response · Authors · 2022-11-17
> **Request for discussion**
>
> Dear Reviewer,
>
> We were wondering if you have gotten the chance to go over our paper updates, and our responses to see if your concerns are resolved. We would be more than happy to resolve any remaining questions you have, and would appreciate it if you engage in a discussion with us.
>
> Thanks!

---

> > ### Author Response · Authors · 2022-11-18
> > **Follow-up**
> >
> > Dear Reviewer,
> >
> > Only a few hours are left in the discussion. We would really appreciate it if you could tell us if your concerns are resolved. We would be more than happy to resolve any remaining questions in the time we have, and would appreciate it if you engage in a discussion with us.

---

> ### Comment · Reviewer_D87g · 2022-11-26
> **Raise my score to 6**
>
> The authors addressed my concerns, and I am happy to raise my rating from 5 to 6.

---

### Official Review · Reviewer_mM3r · 2022-10-24

**Confidence:** 3
**Clarity, Quality, Novelty And Reproducibility:** See above
**Correctness:** 4
**Technical Novelty And Significance:** 3
**Empirical Novelty And Significance:** 3
**Recommendation:** 6

**Strength And Weaknesses:**

## Paper strengths and contributions

**Novelty**
In my opinion, the idea of bridging offline reinforcement learning and behavior cloning to solve novel target tasks in a new domain with only a few demonstrations is intuitive and convincing. This paper presents an effective way to implement this idea.

**Clarity**
The overall writing is clear. The authors utilize figures well to illustrate the ideas.

**Related work**
The authors give a clear description of the related prior works from both the perspectives of offline reinforcement learning and behavior cloning.

**Experimental Results**
The experimental results show that the proposed framework outperforms behavior cloning and offline reinforcement learning baselines including CQL and COG.

## Paper weaknesses and questions

**Experimental results**
- It is mentioned in scenario two that COG does not outperform BC (joint), which is pretty interesting. Unfortunately, COG was only compared to BC but not PTR with this setup. It would be more informative to give a brief explanation or an intuition of why COG was at a disadvantage in that particular scenario.
- It is not entirely intuitive to me why the proposed framework outperforms BC baselines. It would be helpful if the authors give more intuitions.

**Summary Of The Paper:**

This paper addresses the problem of letting robots learn to solve novel target tasks by leveraging diverse offline datasets in combination with only small amounts of task-specific data. To this end, the paper proposes a framework that uses multi-task offline reinforcement learning approaches for pre-training and then fine-tuning its policy on target tasks. The experimental results show that the proposed framework outperforms behavior cloning and offline reinforcement learning baselines including CQL and COG. I am leaning toward accepting this paper since it studies a promising research direction and presents a reasonable framework to address the problem with supporting experimental results.

**Summary Of The Review:**

I am leaning toward accepting this paper since it studies a promising research direction and presents a reasonable framework to address the problem with supporting experimental results.

---

> ### Author Response · Authors · 2022-11-15
> **Author Response (Part 1 of 1): Clarification of COG Baseline and Intuition on why PTR outperforms BC**
>
> Thank you for your constructive feedback and for a positive assessment of this work. We clarify the concerns raised below. We would appreciate it if you could have a look at the responses and let us know if your concerns are resolved.
>
> **We would appreciate it if you were willing to upgrade your rating with the addition of the requested clarifications.**
>
> ____
>
> **Unfortunately, COG was only compared to BC but not PTR with this setup.**
>
> In scenario 2,  we do compare with PTR in scenario 2 as shown in Table 2. Observe that PTR outperforms COG in this setup indicating that pre-training followed by fine-tuning in PTR works better than joint training with COG. We also compare COG and PTR in other scenarios, such as in our simulated experiments (**Table 7**), where we find that PTR again outperforms COG.
>
> ___
>
> **Why COG was at a disadvantage (to BC)  in that particular scenario.**
>
> This is a great question! Unfortunately, we do not have a definitive reason for why BC outperforms COG in scenario 2. One hypothesis is that when performing joint training from the start, the network may choose to represent the bridge data and the target data separately, which could, in turn, hurt performance and limit knowledge transfer between the bridge and target domains. Whereas when performing fine-tuning followed by pre-training, it would be much easier for the training to adapt the pre-trained representation. Though this hypothesis needs further verification and we have now added studying this question as an interesting avenue for future work.
>
> ___
>
> **It is not entirely intuitive to me why the proposed framework outperforms BC baselines.**
>
> This is a great question! Indeed, initially, it was not even intuitive to us why PTR should outperform BC baselines, and hence we performed a controlled empirical analysis to study the same. The discussion titled **“Understanding why PTR outperforms BC baselines”** on Page 8 and its expanded version in **Appendix C.1** attempts to address this. In essence, our observation is that PTR learns to better represent important decision states (such as the time-step before a grasp), thus obtaining a more control-centric representation. In fact, even utilizing the Q-function obtained by PTR to re-weight the data before performing BC leads to substantial performance gains over BC, indicating the advantages of learning a control-centric Q-function.

---

> > ### Comment · Reviewer_mM3r · 2022-11-25
> > **Re: Author Response ...**
> >
> > I appreciate the author's rebuttal, which addresses some of my questions. After carefully reading other reviewers' comments, I have decided to keep my original score.

---

### Official Review · Reviewer_tq7f · 2022-10-31

**Confidence:** 4
**Correctness:** 3
**Technical Novelty And Significance:** 1
**Empirical Novelty And Significance:** 1
**Recommendation:** 3

**Clarity, Quality, Novelty And Reproducibility:**

### Clarity, Quality
The paper is clearly written and easy to follow.

### Novelty
Many aspect of papers can be found in prior works, but it is not enough addressed.

**Strength And Weaknesses:**

### Strength
* Provided results on robotic manipulation tasks can provide empirical knowledge on handling robot control tasks using offline dataset.

### Weakness
* Novelty
    - The problem setup of this paper is very relevant to fully offline meta-RL and the paper is stating "our approach is much more data-efficient and simple". However, any comparison or justification is not given in the paper, thereby misleading readers to be impressed as the problem setup is novel.
    - Given architectural choices are interesting and reasonable, but neither surprising nor significant. Moreover, in order to introduce the network design as a technical novel of this paper, the architecture should be applied and evaluated on more diverse domains.
    - The early stopping criteria is bizarre. The Q-value plot of passed example in Figure 3 is not monotonically increasing over the time. If there is an exact rule deciding whether it is "almost" monotonically increasing it is not described in the main text. If a human decide this, any of given results is reliable as there is no human-related study for this procedure.

* Comparison
    - As mentioned before, the comparison with offline meta-RL is missing. Also, any naive application of imitation learning method other than behavior cloning also deserves to be a baseline.


**Summary Of The Paper:**

This paper targets to solve real-world robotic manipulation tasks by leveraging a large demonstration dataset and a small amount of target task demonstrations.
The proposed method first pre-train a policy on the large dataset with multi-task offline RL, then fine-tune on the small demonstration data again with same objective.
During the fine-tuning phase, a small amount of data is sampled from the dataset used for pre-training to prevent overfitting.
The fine-tuning process is early stopped by heuristic validation rule.
In experimental results, several challenging robotic manipulation tasks are solved by the proposed method using proper design choices of network architecture and network size.

**Summary Of The Review:**

Although this paper provides interesting results on challenging manipulation tasks leveraging offline data,
the relationship between the proposed method and existing methods is not enough addressed.
Moreover, the proposed design choices are incremental but evaluated on a single domain.
Because the generality of the proposed method is not proven, it is difficult to be introduced to the community as a generic learning method for such a problem setup.
Thus, I vote to reject this paper.

---

> ### Author Response · Authors · 2022-11-15
> **Author Response (Part 1 of 2): Added comparison to offline meta-RL, concerns about novelty, number of domains**
>
> Thank you for your detailed feedback. To address your concerns, we have now added additional comparisons to an offline RL method, MACAW (**Table 3**); we clarify that we already have comparisons to more effective imitation methods beyond naive applications of imitation-based methods, and finally clarify our checkpoint selection rule. The changes are shown in $\textcolor{magenta}{magenta}$.
>
> **We would appreciate it if you could have a look at our responses below and the updated version of the paper. Please let us know if this addresses your concerns. We look forward to engaging with you in a discussion.**
>
> ___
>
> **Comparisons:**
>
> To address the concern about comparisons with offline meta-RL methods, we just added a comparison with MACAW (Mitchell et al. ICML 2021), an offline meta-RL algorithm in **Table 3** (details in **Appendix D**). We observe that PTR still outperforms MACAW. MACAW attains a success rate of 0% on all tasks in scenario 3. **PTR outperforms this approach on each task**, averaging a success rate of 48.75%. Qualitatively, we were able to see non-naive behavior such as consistently reaching for the object of interest and lifting it. However, the approach qualitatively was having difficulty with the localization of the target container, thus leading to poor performance. We tuned the hyperparameters for MACAW extensively (see **Appendix D** for details of tuning), but were not able to get a non-zero success rate.
>
> We would also like to clarify that our comparisons do actually already include both **state-of-the-art algorithmic and architectural improvements to BC**, and we find that PTR outperforms all of them. On the architectural front, note that we compare to auto-regressive BC in Table 3. On the algorithmic front, we _already_ compare PTR to behavior transformers (BeT) in Table 3, a recently proposed, state-of-the-art BC method from NeurIPS 2022. We also compare PTR to R3M (Nair et al. 2022) in Table 3, a method that leverages internet-scale video data for learning representations for BC. Finally, we compare PTR against a state-of-the-art self-supervised representation learning approach, MAE (He et al. 2021).
>
> ___
>
> **Novelty:**
>
> We would first emphasize that our work is primarily an application paper, and the main novel contribution is the application of offline RL to pre-train policies for a real-world end-to-end image-based robotic manipulation task. We believe that learning tasks with as few as 10 demonstrations after offline RL pre-training is a substantial novel empirical contribution, and the paper would be of interest to the community because it describes how this novel empirical result can be attained.
>
> We agree that the architectural choices are not novel, but we believe that their use in offline RL is. We believe that this is significant in itself, as no prior work has demonstrated an offline RL approach that can work well for pre-training on **existing, open-source diverse data**, that consists of entirely demonstrations. In fact, prior work (Mandlekar et al. CoRL 2021) only argues that offline RL shouldn’t be useful when provided with human demonstrations, but we show *for the first time* that real-robot offline RL can have benefits compared to imitation.
>
> We believe that the significance of this work is not in the components that we use to construct PTR, but in the result it enabled us to observe for the first time. We strongly believe that this work would be valuable for the robot learning practitioners using offline RL to build upon.
>
> ___
>
> **”Only evaluated on one domain:”**
>
> We evaluate our method on three different toy kitchens as we already show in **Figure 7**, under three different scenarios (Scenarios 1-3), and using two different WidowX robots. As Reviewer D87g also notes, real-robot experiments take a lot of effort to set up: “this is a big plus as such experiments do require a significant amount of effort.” We also believe that the number of domains considered in this paper is substantially larger than many of the other RL and offline RL papers published at robot learning venues.
>
> We are happy to add more toy-kitchen domains, but this would require infrastructure changes (setting up a new toy-kitchen domain, purchasing a new robot), and would not be possible to add in the timeframe of the rebuttal.

---

> > ### Author Response · Authors · 2022-11-15
> > **Author Response (Part 2 of 2): Early stopping criterion**
> >
> > **Early stopping criterion**
> >
> > We have now clarified this detail in **Section 4.2** and **Appendix F**. Like supervised learning, the goal of our early stopping criterion is meant to provide Q-function checkpoints, **within the same run**,  from which the resulting policy is likely to be good. This is a *relative* criterion to be applied within a run. Since the Q-function in CQL is entirely responsible for the policy optimization, _we must make sure that for the chosen checkpoint, the Q-network actually functions like a valid Q-function._
> >
> > One condition that the Q-function must satisfy is that the learned Q-values on state-action pairs in the dataset must increase as a function of time steps within a trajectory. This means that the learned Q-function must show an increasing trend over the course of a trajectory. Of course, this is not exactly possible when faced with errors (and we clarify this in the paper now), but out of all possible checkpoints in a trajectory, it is a perfectly well-defined criterion to pick the checkpoint that is most monotonically increasing.
> >
> > We chose not to utilize a well-defined metric because in our experiments it was pretty clear that there were a few checkpoints much better than others in terms of the relative ordering of the Q-values over time steps in a trajectory. **We detail two worked-out examples in **Appendix F**, where without the need for a formula or a human study, it was evident which checkpoints should be selected.** The same is true for our other experiments.
> >
> > We have added a discussion of how a formula or metric would help in early stopping in general. **Please see Appendix F** for the clarifications.
> >
> > ____
> >
> > **We would appreciate it if you were willing to upgrade your rating with the addition of the requested comparisons and clarifications.**

---

> ### Author Response · Authors · 2022-11-17
> **Request for discussion**
>
> Dear Reviewer,
>
> We were wondering if you have gotten the chance to go over our paper updates, and our responses to see if your concerns are resolved. We would be more than happy to resolve any remaining questions you have, and would appreciate it if you engage in a discussion with us.
>
> Thanks!

---

> > ### Author Response · Authors · 2022-11-18
> > **Follow-up**
> >
> > Dear Reviewer,
> >
> > Only a few hours are left in the discussion. We would really appreciate it if you could tell us if your concerns are resolved. We would be more than happy to resolve any remaining questions in the time we have, and would appreciate it if you engage in a discussion with us.

---

### Decision · Program_Chairs · 2023-01-20

**Decision:**

Reject

**Justification For Why Not Higher Score:**

The main reason for rejecting the paper is the limited technical contribution.

**Justification For Why Not Lower Score:**

N/A

**Metareview: Summary, Strengths And Weaknesses:**

This paper addresses the problem of training real-world robotic manipulation by combining offline datasets and small task-specific datasets. The main contribution of this paper is the experimental work in real world settings, which was valued by all the reviewers. Unfortunately, there is a general concern about the limited technical novelty, and this is the main reason to reject the paper.

As said before, the reviewers strongly value the empirical contribution of this paper and the amount of experiments performed. The reviewers also pointed out the lack of acknowledgement and proper contextualization of the work with respect to previous works. These contextualization aspects were addressed by the authors in their feedback and they also made changes in the paper accordingly. Additionally, some comments related to the need for clarifications on the results  were also addressed by the authors. However, after discussing this paper with the reviewers in a virtual meeting, and also with the Senior AC, we think the strong experimental contribution is not enough for ICLR, since the conference expects contributions with technical novelty. We strongly encourage the authors to submit the work to a robotics conference.


**Summary Of Ac-Reviewer Meeting:**

This work was discussed with the reviewers in a virtual meeting. The reviewers strongly value the empirical contribution of this paper, and the amount of real world experiments done. During the meeting we discussed the limitations of the technical contribution and the fact that the results do not show a clear improvement with respect to previous methods. These two are the main weak points of the paper. During the meeting we also talk about other minor aspects (e.g. the concerns raised by one of the reviewers on the stopping criterion -- the reviewers agreed that stopping criterion is often a weak point, but not a reason to reject the paper). The paper was also discussed with the Senior Area Chair, and unfortunately we all agreed on rejecting the paper due to the limited technical novelty. As said before, we strongly recommend the authors to submit their work to a robotics conference.